# Recruitment of two dyneins to an mRNA-dependent Bicaudal D transport complex

**Thomas E Sladewski[1†‡], Neil Billington[2†], M Yusuf Ali[1†], Carol S Bookwalter[1], Hailong Lu[1], Elena B Krementsova[1], Trina A Schroer[3], Kathleen M Trybus[1]\***

[1]Department of Molecular Physiology and Biophysics, University of Vermont, Burlington, United States; [2]Laboratory of Physiology, National Heart, Lung, and Blood Institute, National Institutes of Health, Bethesda, United States; [3]Department of Biology, Johns Hopkins University, Baltimore, United States

**Abstract** We investigated the role of full-length *Drosophila* Bicaudal D (BicD) binding partners in dynein-dynactin activation for mRNA transport on microtubules. Full-length BicD robustly activated dynein-dynactin motility only when both the mRNA binding protein Egalitarian (Egl) and *K10* mRNA cargo were present, and electron microscopy showed that both Egl and mRNA were needed to disrupt a looped, auto-inhibited BicD conformation. BicD can recruit two dimeric dyneins, resulting in faster speeds and longer runs than with one dynein. Moving complexes predominantly contained two Egl molecules and one *K10* mRNA. This mRNA-bound configuration makes Egl bivalent, likely enhancing its avidity for BicD and thus its ability to disrupt BicD auto-inhibition. Consistent with this idea, artificially dimerized Egl activates dynein-dynactin-BicD in the absence of mRNA. The ability of mRNA cargo to orchestrate the activation of the mRNP (messenger ribonucleotide protein) complex is an elegant way to ensure that only cargo-bound motors are motile.
DOI: https://doi.org/10.7554/eLife.36306.001

**\*For correspondence:**
kathleen.trybus@uvm.edu

[†]These authors contributed equally to this work

**Present address:** [‡]Department of Molecular Microbiology and Immunology, Brown University, Providence, United States

**Competing interests:** The authors declare that no competing interests exist.

## Introduction

Mammalian cytoplasmic dynein-1 (hereafter dynein) is a 12 subunit, 1.4 MDa dimeric molecular motor complex of the AAA+ ATPase family that provides essential cellular functions including transport of vesicles, organelles, and mRNA (reviewed in [*Roberts et al., 2013*]). Dynein is the predominant minus-end directed microtubule-based motor that traffics cellular cargoes for many microns at speeds of ~1 μm/s (*Allan, 2011*). Molecular motors that transport cargo typically exhibit processive behavior when assayed in vitro, meaning that single motors remain bound to the polymer track for long distances without dissociating. It was therefore surprising when in vitro studies showed that single molecules of mammalian dynein were at best weakly processive, even in the presence of the multi-subunit 1.2 MDa dynactin complex that is needed for most cellular functions of dynein (*King and Schroer, 2000*; *McKenney et al., 2014*; *Miura et al., 2010*; *Ross et al., 2006*; *Schlager et al., 2014*; *Trokter et al., 2012*; *Wang et al., 1995*). It was not initially recognized that dynein, like other molecular motors, exists in an auto-inhibited state. One of the first structural studies to investigate dynein auto-inhibition showed that the AAA+ motor domains are closely stacked in a structure called the phi particle (*Amos, 1989*; *Torisawa et al., 2014*). Several non-physiologic mechanisms of disrupting this interaction, such as coupling multiple dyneins to a DNA origami (*Amos, 1989*; *Torisawa et al., 2014*) or binding to a bead (*Belyy et al., 2016*; *King and Schroer, 2000*; *Mallik et al., 2004*; *Nicholas et al., 2015*), converted dynein from a diffusive to a weakly processive motor by separating the stacked rings.

A major advance in understanding dynein function came from single molecule studies, which showed that an α-helical coiled-coil N-terminal fragment of the mammalian adaptor protein Bicaudal-D2 (BicD2N) couples dynein to dynactin. This causes dynein to become highly processive, with

**eLife digest** Cytoplasmic dynein is a motor-like protein that uses energy to transport cargo where it is needed within cells. It moves along protein filaments called microtubules, which act like miniature tracks. Once dynein engages with microtubules, it then picks up cargo using adaptor proteins. In fruit flies, this cargo includes a messenger RNA molecule known as K10, which attaches to dynein via adaptors called Egalitarian and BicD (short for "Bicaudal D"). Egalitarian grabs hold of K10, and BicD links Egalitarian to the dynein motor.

In the absence of a cargo, full-length BicD does not bind to dynein. But, shortening BicD to remove its link to Egalitarian allows it to bind and activate the motor for transport. Much of our current understanding of dynein comes from studies that use shortened adaptor proteins like these. These proteins cannot bind cargo, so we know little about how the cargo and the adaptors control dynein activity.

To address this, Sladewski, Billington, Ali et al. purified the components of the K10 transport system and then recreated it in the laboratory. This revealed that it is BicD that decides when dynein is ready to go. First, imaging techniques showed that empty BicD forms a looped shape that hides the part of its structure that binds to dynein. This essentially switches it "off", preventing empty dynein motors from moving. When the K10 cargo is ready for transport, it binds to two Egalitarian molecules, which work together to uncurl the BicD loop. This frees up the end of the BicD molecule, allowing it to link up with the dynein motor.

The key to uncurling the BicD protein was the presence of two Egalitarian molecules. And it was the cargo, K10, that brought them together, ensuring that the motors only moved when the cargo was ready. What is more, the uncurled BicD could bind not one but two dyneins. This allowed the cargo to move faster, and over longer distances than cargo with one dynein motor.

Recreating molecular machines and imaging their molecules provides a way to understand how they work. Studying how dynein moves cargo is key to understanding how molecules are transported within cells. This, in turn, could reveal what happens when the system goes wrong. Transport defects can cause diseases in humans, including neurodegenerative diseases. As such, a better understanding of how the transport system works may one day open new avenues for health research.

DOI: https://doi.org/10.7554/eLife.36306.002

5–10 μm run lengths, and leads to motor accumulation at the microtubule minus-end (*McKenney et al., 2014*; *Schlager et al., 2014*). This minimal tripartite complex is called DD (BicD2N) (dynein-dynactin-BicD2N). Dynein in the activated DD(BicD2N) complex produced 4.3 pN of force (*Belyy et al., 2016*), considerably higher than the 0.5–1.5 pN forces reported earlier for dynein alone (*Mallik et al., 2004*; *McKenney et al., 2010*; *Ori-McKenney et al., 2010*; *Rai et al., 2013*), thus allowing dynein to successfully engage in a tug-of-war with a single kinesin. Recent EM studies provided further insight into the structural basis for the inhibition and activation of dynein. Cryo-EM studies showed that when the dynein motor domains are stacked in the phi particle they are locked in a conformation with low affinity for microtubules; disruption of the motor domain self-dimer creates an 'open' state that is still inhibited for motion (*Zhang et al., 2017*). Only when bound to dynactin and an 'adaptor' such as BicD2N are the dynein motor domains aligned in a parallel orientation on the microtubule that correlates with highly processive movement (*Chowdhury et al., 2015*).

Cargo adaptor proteins such as BicD2 are thus central to control of dynein activity in the cell. Although BicD2N artificially fused to mitochondria or peroxisome targeting sequences supports robust dynein-driven motility in HeLa cells (*Hoogenraad et al., 2003*), full-length BicD has only a mild effect on organelle re-localization (*Hoogenraad et al., 2003*). It has been assumed that full length BicD2 assumes an auto-inhibited conformation that does not bind to or activate dynein-dynactin. BicD2 is composed of three α-helical coiled-coil domains; the N-terminal domain (CC1) is involved with dynein-dynactin binding and activation (*Urnavicius et al., 2015*), whereas the C-terminal domain (CC3) binds adaptor proteins that link dynein to cargo (*Liu et al., 2013*). Early yeast-2-hybrid studies showed that the CC1 domain interacts with the CC3 domain, leading to a model in

which BicD forms N- to C-terminal auto-inhibitory interactions (*Hoogenraad et al., 2001*). Early metal-shadowed images of *Drosophila* BicD, in contrast, suggested interaction of CC3 with CC2 (*Stuurman et al., 1999*). The leading hypothesis for cellular activation of dynein-dynactin transport by BicD is that cargo-associated BicD2-binding proteins relieve auto-inhibition by freeing up CC1.

Here, we test this cargo-activation model by reconstituting in vitro a biologically well-characterized system, localization of *K10* mRNA in *Drosophila*. During development, *K10* mRNA is transported by dynein from nurse cells to the *Drosophila* oocyte, where it localizes to the anterior margin to establish the dorso-ventral axis of the *Drosophila* egg (*Cheung et al., 1992*). *K10* contains a single 44 base-pair transport/localization element (TLS) in its 3'UTR that binds the mRNA-binding adaptor protein Egalitarian (Egl), which in turn binds to CC3 of *Drosophila* BicD (*Dienstbier et al., 2009*; *Serano and Cohen, 1995*).

To understand the molecular basis for dynein activation in this system, we reconstituted a motile mRNP (messenger ribonucleotide protein) complex in vitro from purified dynein, dynactin, full-length BicD, Egl, and synthesized *K10* mRNA. Single-molecule approaches coupled with electron microscopy suggest that the key to dynein activation is disruption of an auto-inhibited loop structure of BicD. A single *K10* mRNA can bind two Egl molecules to create a bivalent Egl that efficiently disrupts the BicD loop structure, freeing CC1 and favoring recruitment of two dynein motors to the complex. The resulting mRNPs move at higher speeds and over longer distances than complexes with just one dynein. *K10* mRNA cargo thus orchestrates activation of the motile complex, an elegant mechanism ensuring that only motors properly complexed with cargo can undergo robust motility.

## Results

### Full-length BicD forms an auto-inhibited looped conformation that does not bind dynein-dynactin

A single-molecule pulldown assay was used to determine if full-length *Drosophila* BicD (hereafter called BicD) binds to dynein-dynactin purified from bovine brain. BicD is composed of predicted α-helical coiled-coil domains (CC1, CC2, and CC3, colored bars) separated by non-coiled coil regions (gray bars) (*Figure 1A*). As positive controls, we tested the binding of a truncated version of *Drosophila* BicD (hereafter called BicD[CC1]), as well as a truncated version of mammalian BicD2 (hereafter called BicD2N) (see *Figure 1—figure supplement 1* for SDS-PAGE). These same N-terminal fragments of *Drosophila* BicD (*Dienstbier et al., 2009*) and mammalian BicD2 (*Hoogenraad et al., 2003*) stimulate dynein-based transport when exogenously expressed in cultured cells, and BicD2N has been shown to couple dynein to dynactin and highly activate processive motility in vitro (*McKenney et al., 2014*; *Schlager et al., 2014*).

Full-length and truncated Qdot labeled-BicD constructs were incubated with dynein and dynactin, and applied to flow cells with surface-adhered microtubules in the presence of the ATP analog AMP-PNP, which causes dynein to bind strongly to microtubules. Total internal reflection fluorescence (TIRF) microscopy was used to visualize and quantify associations with the microtubule (*Figure 1B,C*). Binding of BicD or BicD[CC1] alone to microtubules was negligible (*Figure 1B,C*), and thus BicD recruitment to microtubules is an accurate reporter of formation of a tripartite BicD-dynein-dynactin complex. BicD[CC1] showed an ~10 fold enhanced recruitment to microtubule-bound dynein-dynactin compared with full-length BicD, similar to the ~13 fold increase observed when mammalian BicD2N was compared with full-length BicD (*Figure 1D*). The truncated versions of *Drosophila* BicD and mammalian BicD2 thus behave similarly, suggesting that their mechanism of action is evolutionarily conserved.

Electron microscopy was used to determine the structural basis for BicD auto-inhibition. Negatively stained EM images of YFP-BicD show two distinct globular densities at the N-terminus that correspond to YFP, confirming the formation of a parallel coiled-coil dimer. The montage (*Figure 2A*) is arranged to show (row 1) the most common 'b' orientation molecules, (row 2) a less common 'd' orientation, (row 3) the range over which the molecule can flex, (row 4) some very compact molecules, and (row 5) rare open molecules. The most common feature of BicD is the loop which appears to be formed by parts of all three coiled coil segments, which is readily seen in averages of all molecules ('global') as well as averages of the most common 'b' orientation molecules (*Figure 2B,C*, *Figure 2—figure supplement 1*).

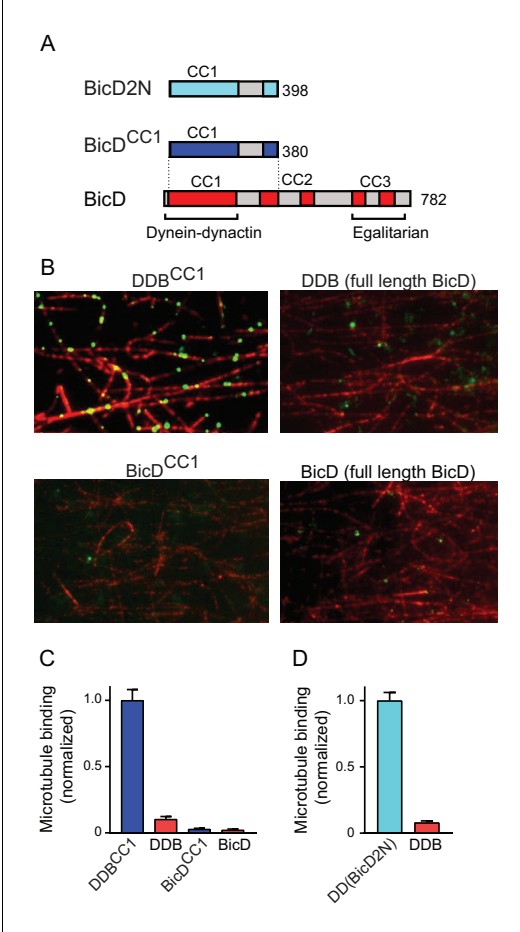

**Figure 1.** Full-length BicD is auto-inhibited and does not bind dynein-dynactin. (**A**) Schematic of BicD constructs. Full-length *Drosophila* BicD contains three coiled-coil (CC) regions that are designated CC1, CC2, and CC3 (red). CC2 and CC3 are each interrupted by non-coiled-coil regions (gray). Regions predicted to form coiled-coil domains were determined from Paircoil2 analysis (*McDonnell et al., 2006*). Truncated versions of *Drosophila* BicD (BicD^CC1) and mammalian BicD2 (BicD2N) are also shown. CC1 is the minimal fragment that activates dynein-dynactin, but does not bind cargo adaptor proteins, which occurs via CC3. (**B**) Single-molecule pulldowns on microtubules (red) show that dynein-dynactin associates with BicD^CC1 (green) but not with full-length *Drosophila* BicD (green) in the presence of AMP-PNP (upper panels). Controls show that the two BicD constructs alone do not bind non-specifically to microtubules (lower panels). BicD was visualized with a 525 nm-streptavidin Qdot bound to an N-terminal biotin tag on BicD. (**C**) Quantification of the number of dynein-dependent associations of complexes containing *Drosophila* BicD^CC1 (blue) versus full-length *Drosophila* BicD (red) to microtubules, normalized to microtubule length and dynein concentration (average ± SEM from 5 to 15 fields). Controls for non-specific binding are also quantified.

The YFP densities abut a fairly straight CC1 region followed by a flexible break in the coiled-coil from which CC2 and CC3 loop back to interact at a point approximately halfway along the CC1 coiled coil (*Figure 2D,E*). Our interpretation that the relatively straight portion of BicD is CC1 is consistent with contour length measurements. The distance from YFP to the very bottom of the loop is 38.7 ± 3.8 nm (± SD, n = 150), in good agreement with Paircoil2 analysis (*McDonnell et al., 2006*) that predicts the first 258 amino acids of BicD (CC1) to be an α-helical coiled-coil 37.7 nm long (based on 0.146 nm rise/residue). The contour length from YFP to the point where the CC2-CC3 loop contacts CC1 is 16.5 ± 2.3 nm (± SD, n = 133), and from here to the bottom of the loop is 19.1 ± 2.0 nm (± SD, n = 150). The sum of these lengths (16.5 + 19.1 nm) is 35.6 nm, similar to the 38.7 nm measured from YFP to the base of the loop. The contour length of the curved side of the loop is 24.5 ± 2.7 nm (± SD, n = 148). Together, these measurements predict that CC2-CC3 loop contacts CC1 near amino acid 113 (16.5 nm/0.146 nm rise per residue).

Pulldowns confirmed that the CC1 domain interacts with a construct containing the CC2-CC3 domains, but not with a construct composed only of CC2 (*Figure 2F*), consistent with our interpretation of the EM images.

## Binding of Egl does not disrupt the auto-inhibited structure of BicD

BicD and the mRNA-binding protein Egl can either be expressed separately and then reconstituted after purification, or co-expressed as a complex in Sf9 cells (*Figure 3—figure supplement 1*). Electron microscopy of the co-expressed BicD-Egl complex revealed a remarkably similar overall conformation to that of BicD alone, with the loop remaining intact, and the paired N-terminal YFP domains confirming the BicD parallel coiled-coil (*Figure 3A,C–F*). An additional globular structure was often observed adjacent to the loop (*Figure 3A,F*, *Figure 3—figure supplement 4*). The N-terminal region of Egl (amino acids 1–79) binds BicD, and residues 557–726 of Egl contain a globular exonuclease homology region (*Dienstbier et al., 2009*) that may be the globular structure we observe. Averages of molecules in the 'b' orientation (*Figure 3—figure supplement 4*) are very similar to those seen in the absence of Egl (*Figure 2—figure supplement 1*). Co-alignment and classification of BicD-Egl images with images of BicD alone revealed that the most consistent difference in density lies

*Figure 1 continued*

(D) Quantification of the number of dynein-dependent associations of complexes containing mammalian BicD2N (cyan) or full-length *Drosophila* BicD (red) to microtubules, normalized to microtubule length and dynein concentration (average ± SEM from 5 to 15 fields), from independent experiments. See *Figure 1— source data 1*.

DOI: https://doi.org/10.7554/eLife.36306.003

The following source data and figure supplement are available for figure 1:

**Source data 1.** Dataset for *Figure 1*.

DOI: https://doi.org/10.7554/eLife.36306.005

**Figure supplement 1.** SDS-PAGE gels of BicD constructs and tissue-purified bovine brain dynein and dynactin.

DOI: https://doi.org/10.7554/eLife.36306.004

near the bottom center of the loop, suggesting that this region is where Egl is bound (*Figure 3G*, *Figure 3—figure supplement 5F*). Some raw images suggest attachment to BicD near to the top of the loop, but this was not evident in the difference map of the aligned images. Classification using a mask adjacent to the loop intended to highlight the position of the Egl globular domain revealed it can occupy a range of positions, consistent with single-particle images (*Figure 3F*, *Figure 3—figure supplement 4*). Positional variability can also be seen in a heatmap illustrating the location of the globular domain in aligned images (*Figure 3H*).

Taken together, the EM data suggest the presence of a long flexible linker between the N-terminal BicD-binding region of Egl and the globular domain. A potential complication is the presence of the YFP domains at the N-terminus of CC1, which is itself flexible. To rule out that the additional density corresponds to the YFPs, we aligned and classified images of Egl bound to BicD lacking YFP. The appearance was strikingly similar to that of YFP-BicD-Egl but the two large YFP domains were no longer present. By contrast, the globular domain adjacent to the loop was still observed (*Figure 3—figure supplement 2C*). This verified the assignment of the YFP domains and further demonstrated that the BicD structure is not significantly affected by the YFP tag. Alignment and classification of this complex gave very similar results to those obtained for YFP-BicD-Egl, revealing once again the characteristic loop structure and adjacent globular domain (*Figure 3—figure supplement 5A–C*). *Video 1* shows a low-resolution 3D map of negative stain EM data, which allows size comparison of the loop with existing structures for parts of the YFP-BicD-Egl complex.

The EM images imply that Egl alone is not sufficient to disrupt the auto-inhibited conformation of BicD. This result predicts that the BicD-Egl complex will not bind to dynein-dynactin. Single-molecule pulldowns confirmed that full-length YFP-BicD and Egl-Qdot were not recruited to microtubule-bound dynein-dynactin (DDBE) (*Figure 4A,B*).

## The BicD-Egl complex binds and robustly activates dynein-dynactin in the presence of mRNA

We next tested by single-molecule pulldowns whether mRNA cargo was required for BicD-Egl to bind to dynein-dynactin. Dynein-dynactin, BicD, Egl (DDBE) plus *K10* mRNA was applied to surface-adhered microtubules in the presence of AMP-PNP to visualize complexes associated with bound dynein-dynactin. The presence of *K10* mRNA enhanced Egl and BicD colocalization with microtubule-bound dynein-dynactin (*Figure 4A,B*). This observation implies that both mRNA and Egl are needed for BicD to adopt a conformation that can recruit dynein-dynactin. Consistent with this, electron microscopy of the YFP-BicD-Egl-*K10*$_{min}$ mRNA complex showed a variety of flexible structures, with the auto-inhibited loop conformation seen for YFP-BicD alone or YFP-BicD-Egl largely absent (*Figure 3B* vs. *Figure 3A*). The same result was obtained with the construct lacking YFP (*Figure 3— figure supplement 2C* vs. *Figure 3—figure supplement 2D*). The highly variable appearance of the BicD-Egl-mRNA complexes hampered image alignment and resulted in heterogeneous class averages with insufficient features for detailed assignment of the individual components (*Figure 3—figure supplement 5D,E*). A small number of particles (<10%) of YFP-BicD-Egl-mRNA produced classes resembling the looped structure. It is unclear if this represents incomplete occupancy of Egl with mRNA, or whether BicD retains a weak propensity for auto-inhibition even in the presence of Egl-mRNA. We also examined YFP-BicD in the presence of mRNA and saw no disruption of the loop structure or binding of mRNA to the loop, indicating that only the combination of Egl and mRNA is sufficient to relieve BicD auto-inhibition (*Figure 3—figure supplement 2B*).

Single-molecule pulldowns were further used to assess the requirement for a zip code in *K10* mRNA for dynein-dynactin recruitment by BicD-Egl-mRNA complexes. mRNPs reconstituted with a

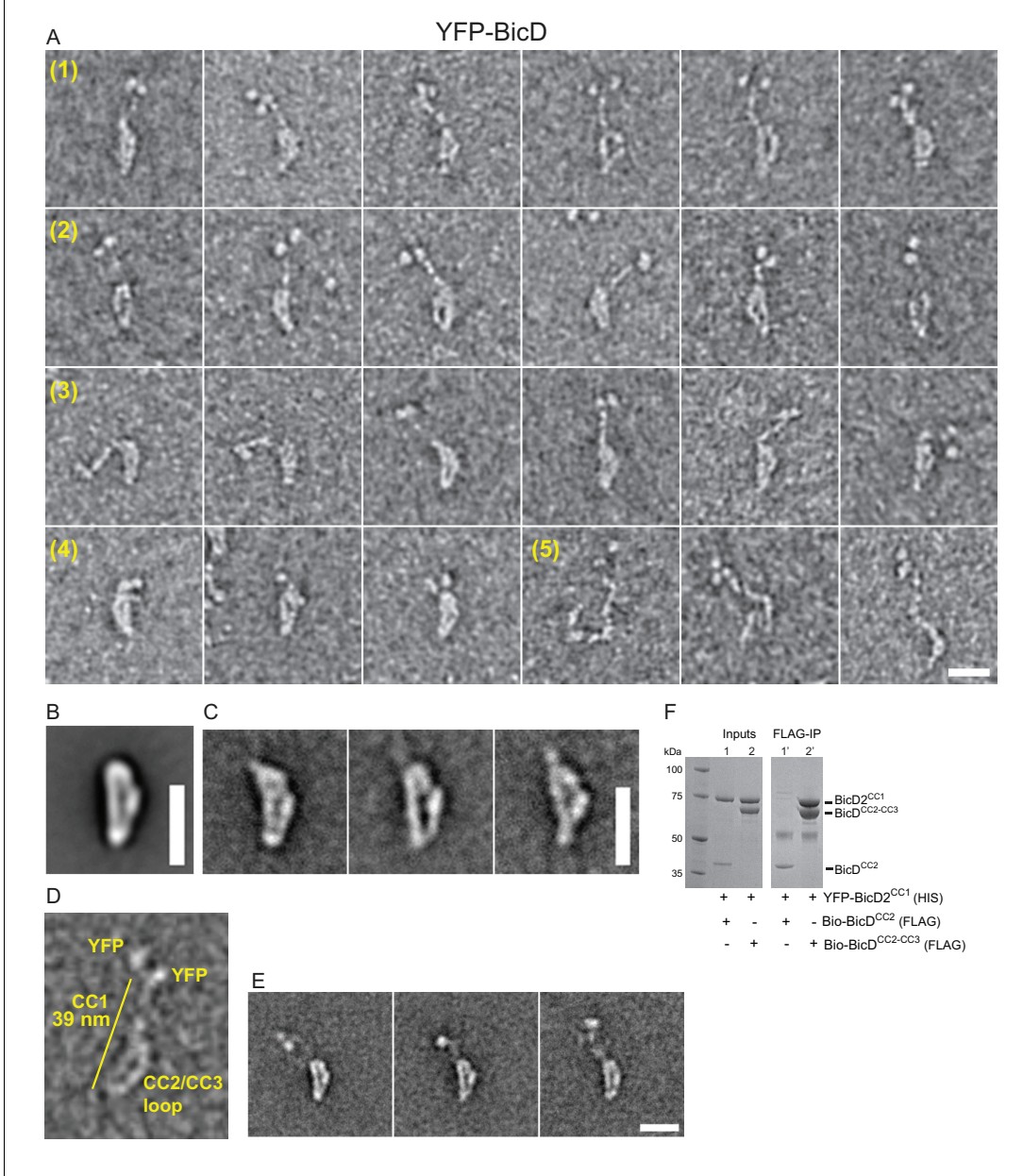

**Figure 2.** Electron micrographs of the auto-inhibited full-length YFP-BicD. (**A**) A montage of negatively stained images showing (row 1) the most common 'b' orientation, (row 2) a less common 'd' orientation, (row 3) the range over which the molecule can flex, (row 4) compact molecules, and (row 5) rare open molecules. (**B**) Global average of all aligned YFP-BicD images. (**C**) Example class averages of BicD with classification focused on the loop. (**D**) One YFP-BicD molecule illustrating the interpretation of the EM image, with the length of the long straight section indicated. (**E**) Example class averages of YFP-BicD with classification focused on the protruding YFP-CC1 region. See **Figure 2—figure supplement 1** for full classifications. (**F**) Pulldown showing that CC1 interacts with CC2-CC3 but not CC2. Scale bars = 20 nm.

DOI: https://doi.org/10.7554/eLife.36306.006

The following figure supplement is available for figure 2:

**Figure supplement 1.** Image processing scheme for YFP-BicD EM data.

DOI: https://doi.org/10.7554/eLife.36306.007

mutant *K10* mRNA transcript lacking the transport/localization sequence (TLS) caused a ~2-fold reduction in the number of BicD-Egl complexes associated with microtubule-bound dynein-dynactin, compared with native *K10* (**Figure 4A,B**). This suggests that Egl binds primarily to the TLS zip code but is also able to bind weakly to other regions of the mRNA.

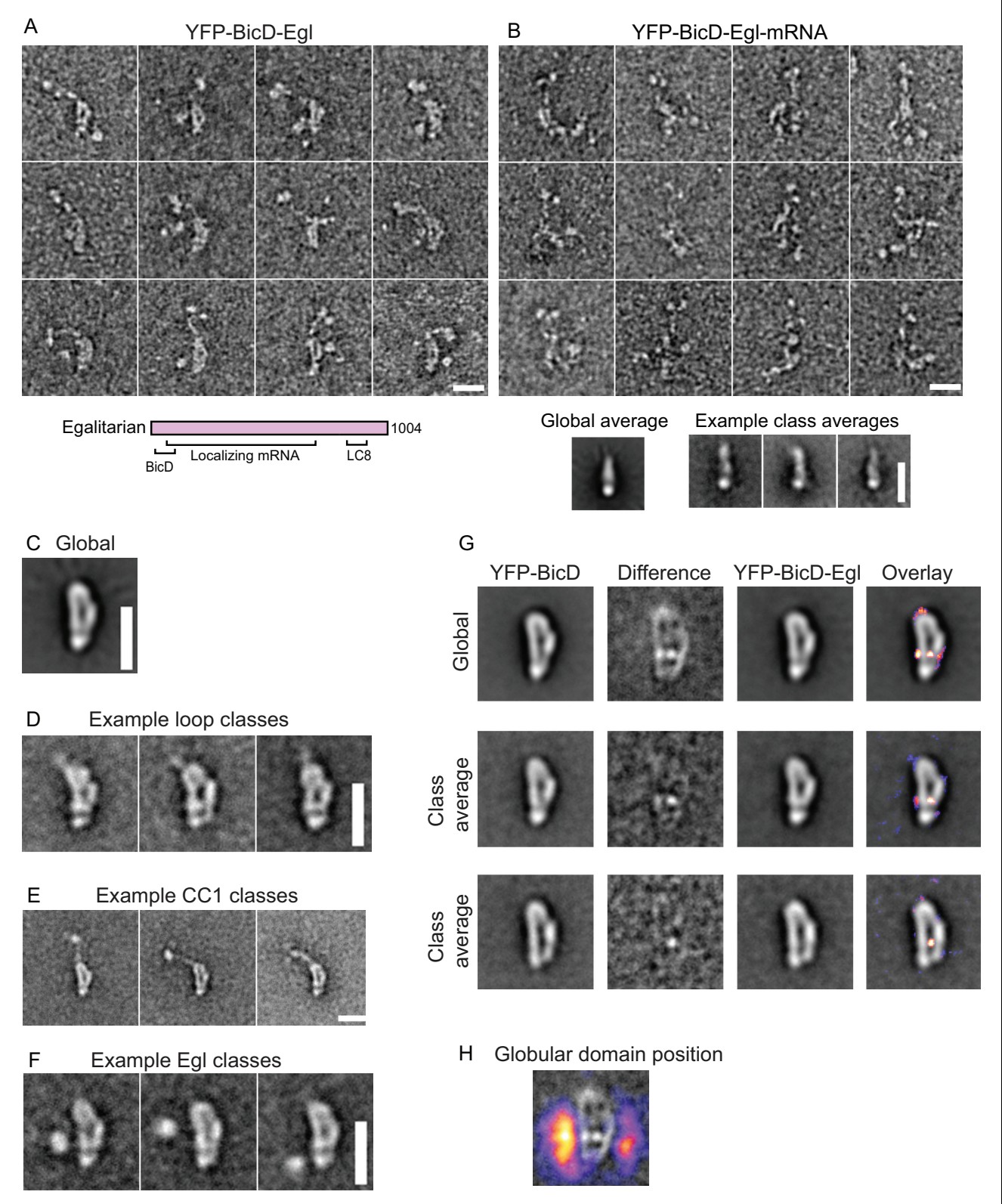

**Figure 3.** Electron micrographs of the YFP-BicD-Egl and YFP-BicD-Egl-mRNA complex. (**A**) Montage of images showing that BicD retains the auto-inhibited looped conformation in the presence of bound Egl. A schematic of Egl (below) shows that the N-terminal domain binds BicD, the C-terminus binds dynein light chain (LC8), and mRNA binding is mediated through a large number (~800) of amino acids. (**B**) Montage of images showing that YFP-BicD no longer retains the auto-inhibited looped conformation in the presence of bound Egl and mRNA ($K10_{min}$). Global and class averages are shown

*Figure 3 continued on next page*

*Figure 3 continued*

below the montage. (C) Average of all YFP-BicD-Egl molecules. (D) Example class averages of YFP-BicD-Egl with classification focused on the loop. (E) Example class averages of YFP-BicD-Egl with classification focused on the protruding YFP-CC1 region. (F) Example class averages of YFP-BicD-Egl with classification focused adjacent to the loop, to reveal the globular domain of Egl. (G) Co-alignment and classification of YFP-BicD and YFP-BicD-Egl. Upper row shows the average of YFP-BicD and YFP-BicD-Egl images (after an initial alignment, classification and selection of particles in similar orientations). Difference maps are shown in the second column and the difference map is overlaid as a heatmap onto the average of YFP-BicD-Egl in the fourth column, to show clearly the locations of greatest difference. Second and third rows show the averages and differences for individual classes. See *Figure 3—figure supplement 5* for the full classification. (H) Heatmap showing the position of the globular Egl domain. Scale bars = 20 nm.

DOI: https://doi.org/10.7554/eLife.36306.008

The following figure supplements are available for figure 3:

**Figure supplement 1.** SDS-PAGE gels of Egl.
DOI: https://doi.org/10.7554/eLife.36306.009

**Figure supplement 2.** EM of other conditions imaged in this study.
DOI: https://doi.org/10.7554/eLife.36306.010

**Figure supplement 3.** Fields of view of all conditions imaged in this study.
DOI: https://doi.org/10.7554/eLife.36306.011

**Figure supplement 4.** Image processing scheme for YFP-BicD-Egl EM data.
DOI: https://doi.org/10.7554/eLife.36306.012

**Figure supplement 5.** Other classifications used in this study.
DOI: https://doi.org/10.7554/eLife.36306.013

The run frequencies of the reconstituted complexes were assayed by TIRF microscopy using either Qdot-labeled adaptor proteins (BicD or Egl), or *K10* mRNA synthesized with Alexa Fluor 488-UTP. In the absence of either BicD or Egl, minimal movement of labeled *K10* mRNA was observed (*Figure 4C*). In the presence of both BicD and Egl but absence of mRNA, a low-run frequency (20% of maximal) was observed. Inclusion of *K10* mRNA enhanced run frequency five-fold. The effect of addition of mRNA cargo is illustrated in kymographs (*Figure 4D*). As a final control, we confirmed that *K10* mRNA and Egl colocalize in motile mRNP complexes (*Figure 4E*).

Similar to the single-molecule pulldowns, *K10* mRNA constructs lacking the TLS zip code showed a reduced run frequency (35% of maximal) compared with wild-type *K10* transcripts. A similar low frequency was observed with two heterologous mRNAs, mammalian β-actin and *S. cerevisiae ASH1* mRNA. Robust cellular localization of *K10* mRNA requires the TLS zip code (*Bullock and Ish-Horowicz, 2001*; *Serano and Cohen, 1995*), but even non-localizing mRNAs associate with dynein and move bidirectionally in *Drosophila* embryos (*Amrute-Nayak and Bullock, 2012*).

## *K10* mRNA activates mRNPs for fast, processive movement

The motile properties of the fully reconstituted mRNPs (dynein-dynactin-BicD-Egl-*K10* mRNA) were compared with the minimal complex

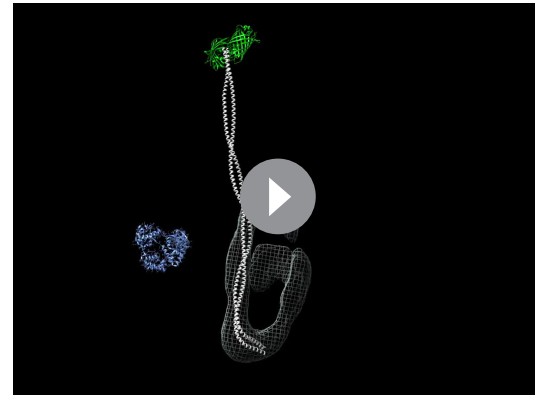

**Video 1.** Low-resolution 3D map of negative stain EM data (related to *Figure 3*). The video shows the size comparison of the apparent loop (EM volume depicted in gray mesh) with existing structures for parts of the YFP-BicD-Egl complex. The map has not been validated, and serves only as a guide. YFP molecules at the N-terminus of BicD are depicted in green (PDB ID: 2Y0G). The blue molecule is the exonuclease domain of RNAseD (PDB ID:1YT3), which serves as a proxy for the proposed exonuclease-like domain of Egl (*Mach and Lehmann, 1997*; *Moser et al., 1997*; *Navarro et al., 2004*). The white coiled-coil is the BicD2 N-terminal fragment from the structure of the dynein-dynactin-BicD2 complex (*Urnavicius et al., 2015*) (PDB ID:5AFU), fitted into the straighter part of the looped structure. In this interpretation, the remaining loop bulge must consist of CC2 and CC3. Given that CC2 is contiguous in sequence with the end of CC1, CC2 is likely to be the bottom part of the 'b' structure, while CC3 binds to a region near the middle of CC1.

DOI: https://doi.org/10.7554/eLife.36306.014

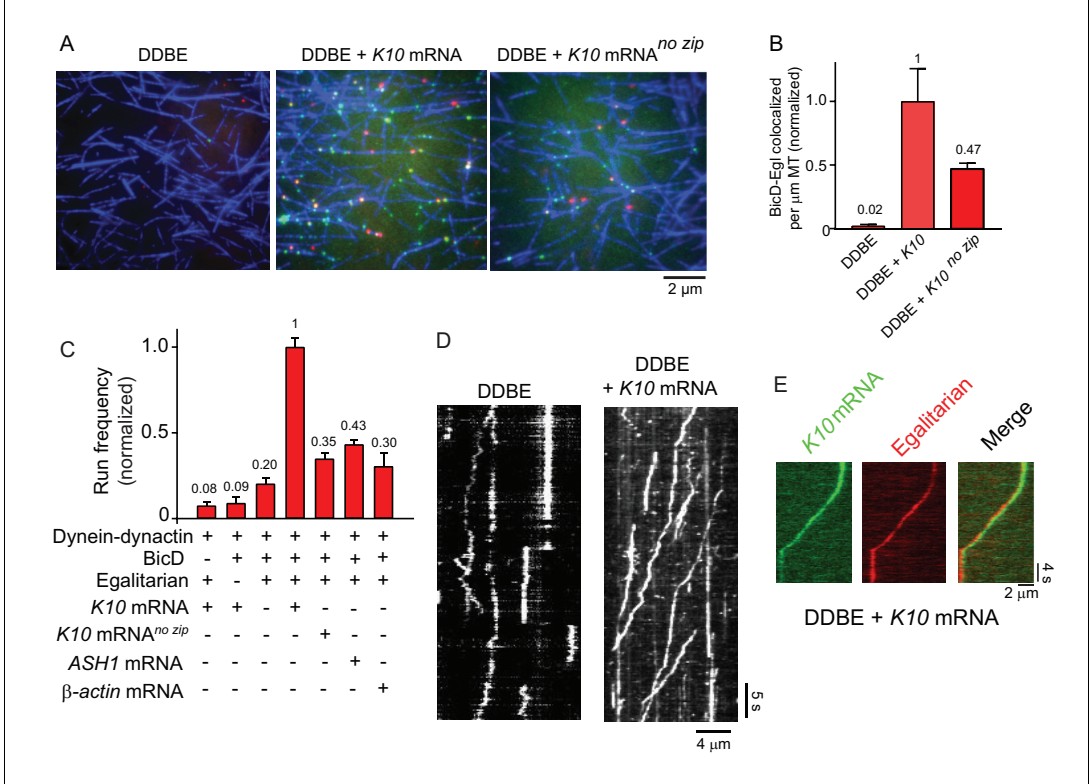

**Figure 4.** *K10* mRNA is needed for BicD-Egl to recruit dynein-dynactin for motility. (**A**) Single-molecule pulldowns of YFP-BicD (green) and Qdot-Egl (red) by dynein-dynactin (DDBE) bound to microtubules (blue) in the absence or presence of *K10* mRNA. (**B**) Quantification of the single-molecule pulldown data showing the normalized number of colocalized BicD-Egl complexes bound to dynein-dynactin per µm microtubule length in the absence of mRNA (DDBE), in the presence of *K10* mRNA (DDBE + *K10*), or *K10* without the TLS zip code (DDBE + *K10* [no zip]). (**C**) Normalized run frequencies (per µM dynein per µm microtubule length per time) of motile mRNPs. (Bars, left to right) Little movement of labeled *K10* mRNA in the absence of either BicD or Egl. Few events are also observed for mRNPs lacking mRNA (movement was visualized with a Qdot bound to Egl). Fully reconstituted mRNPs with *K10* mRNA have the highest run frequency, while mRNPs reconstituted with *K10* mRNA lacking the TLS zip code showed reduced run frequencies. Two unrelated mRNA constructs, *S. cerevisiae ASH1* mRNA and mammalian β-actin mRNA showed run frequencies similar to that of *K10* mRNA lacking the TLS zip code. Error bars, sem, n ≥ 4 movies per condition, two independent experiments. (**D**) Kymographs illustrating motion of dynein-dynactin-BicD-Egl (DDBE) in the absence or presence of *K10* mRNA. (**E**) Kymograph showing moving DDBE-*K10* mRNA complexes dual labeled with Egl bound to a Qdot (red) and *K10* mRNA labeled with Alexa Fluor 488-UTP (green). See *Figure 4—source data 1*.

DOI: https://doi.org/10.7554/eLife.36306.015

The following source data is available for figure 4:

**Source data 1.** Dataset for *Figure 4*.

DOI: https://doi.org/10.7554/eLife.36306.016

containing dynein-dynactin-BicD[CC1], and in separate experiments with the previously characterized dynein-dynactin-BicD2N complex (*McKenney et al., 2014*; *Schlager et al., 2014*). Kymographs demonstrating the observed motility are shown in *Figure 5A* and *Figure 5—figure supplement 1*.

The speed of *K10* mRNPs was not significantly different than the minimal DDB[CC1] complex (0.42 ± 0.22 µm/s, n = 505 vs. 0.43 ± 0.26 µm/s, n = 516; p=0.25, t-test, mean ± SD) (*Figure 5B*). *K10* mRNP run lengths were also similar to the minimal DDB[CC1] complex (6.4 ± 0.25 µm, n = 71 vs. 5.3 ± 0.17 µm, n = 66; p=0.8, Kolmogorov–Smirnov test, mean ± SE) (*Figure 5C*).

When the fully reconstituted mRNP was independently compared with dynein-dynactin-BicD2N, *K10* mRNP motility speed was faster than the minimal DDB[CC1] complex (0.45 ± 0.21 µm/s, n = 1126 vs. 0.35 ± 0.19 µm/s, n = 1147, p<0.05, t-test, mean ± SD) (*Figure 5D*). Longer run lengths for *K10* mRNP were also observed (7.2 ± 0.5 µm, n = 142 vs. 5.4 ± 0.7 µm, n = 137; p=0.053, Kolmogorov–Smirnov test, mean ± SE) (*Figure 5E*).

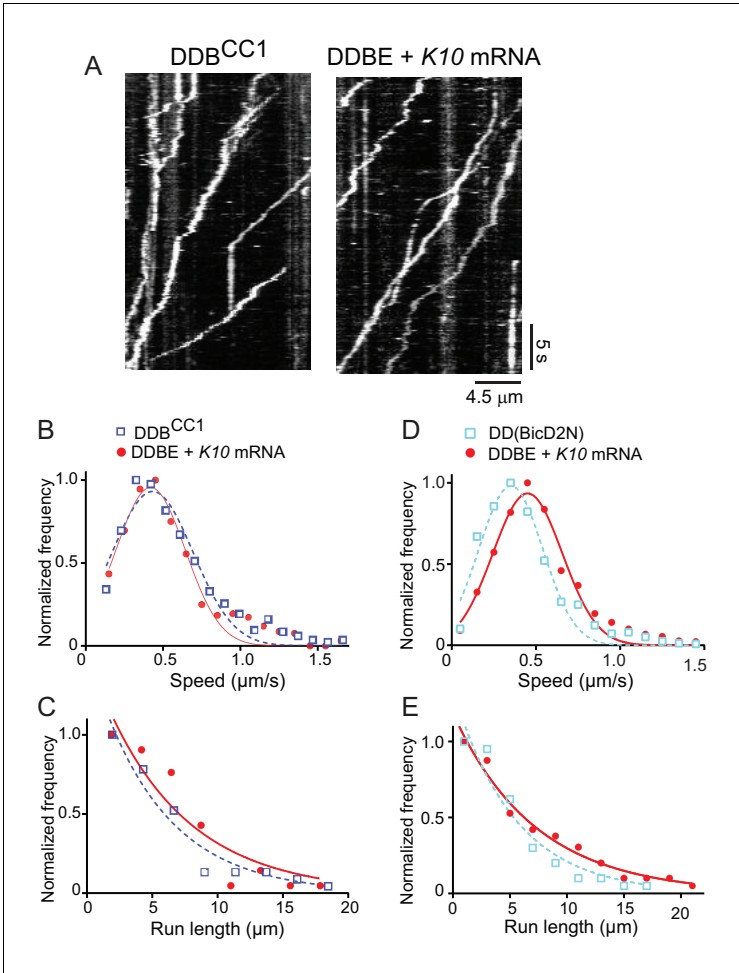

**Figure 5.** The motile properties of the fully reconstituted mRNP are similar to dynein-dynactin complexes reconstituted with Drosophila DDB$^{CC1}$ or mammalian BicD2N. (**A**) Kymograph of (left panel) a minimal dynein-dynactin-BicD$^{CC1}$ (DDB$^{CC1}$) complex, and (right panel) a complex of dynein, dynactin, full-length *Drosophila* BicD, Egl and *K10* mRNA, visualized with a Qdot on BicD (**B**) Speed distributions of DDBE + *K10* mRNA (red circles) (0.42 ± 0.22 µm/s, n = 505) and DDB$^{CC1}$ (blue squares) (0.43 ± 0.26 µm/s, n = 516, p=0.25, t-test, mean ± SD). (**C**) Run length distributions of DDBE + *K10* mRNA (red circles) (6.4 ± 0.25 µm, n = 71) and DDB$^{CC1}$ (blue squares) (5.3 ± 0.17 µm, n = 66, p=0.8, Kolmogorov–Smirnov test, mean ±SE of the fit). (**D**) Speed distributions of DDBE + *K10* mRNA (red circles) (0.45 ± 0.21 µm/s, n = 1126) and DD(BicD2N) (blue squares) (0.35 ± 0.19 µm/s, n = 1147; p<0.05, t-test, from 50 representative run trajectories for each condition, mean ± SD). (**E**) Run length distributions of DDBE + *K10* mRNA (red circles) (7.2 ± 0.5 µm, n = 142) and DD (BicD2N) (blue squares) (5.4 ± 0.7 µm, n = 137, p=0.053, Kolmogorov–Smirnov test, mean ± SE of the fit). See *Figure 5—source data 1*.

DOI: https://doi.org/10.7554/eLife.36306.017

The following source data and figure supplement are available for figure 5:

**Source data 1.** Dataset for *Figure 5*.
DOI: https://doi.org/10.7554/eLife.36306.019

**Figure supplement 1.** Kymograph of (left panel) a minimal dynein-dynactin-BicD2N complex, visualized with a Qdot bound to BicD2N, and (right panel), a complex of dynein, dynactin, full-length *Drosophila* BicD, Egl and *K10* mRNA labeled with Alexa Fluor 488-UTP.
DOI: https://doi.org/10.7554/eLife.36306.018

Despite minor differences, all three complexes show long processive runs of ~5–7 µm at speeds of ~0.4 µm/s (*Figure 5B–E*). On this basis, we conclude that the fully reconstituted mRNP is as active as the minimal DDB$^{CC1}$ or DD(BicD2N) complex.

## Motile mRNPs contain two copies of Egl

BicD is a coiled-coil dimer, and thus has the potential to bind two molecules of Egl. To determine the stoichiometry of Egl binding and test whether mRNP speed and run length is affected by Egl stoichiometry, we prepared Egl labeled with either a green (565 nm) or a red (655 nm) streptavidin-Qdot via a C-terminal biotin tag. The samples were blocked with excess biotin to prevent additional Qdot binding. *K10* mRNPs were then reconstituted with equimolar red- and green-labeled Egl (*Figure 6A*). Motile complexes containing both green and red Qdots were observed, as were motile complexes with a single color (*Figure 6B*). Single-colored complexes can contain either one or two Egls of the same color. After correction for complexes with two copies of the same color Egl (see Materials and methods), 86.7% percent of complexes were determined to contain two molecules of Egl. In rare cases, a moving complex started dual-colored, and then reduced to a single color in the same trajectory, indicating that Egl dissociation does not immediately terminate motion.

The speed and run length distributions of the single versus dual-colored complexes differed in an interesting way. The speed histogram of the dual-colored complex was best fit with a double Gaussian distribution with a slow speed of 0.27 ± 0.10 µm/s, and a faster speed of 0.73 ± 0.16 µm/s (n = 62). Speeds of the single-colored complexes were better fit to a broad single Gaussian distribution, with a speed of 0.38 ± 0.20 µm/s (n = 81) (*Figure 6C*). Run length distributions of the single-colored runs were significantly shorter than the dual-colored runs (6.4 ± 0.7 µm, n = 81 vs.12.1 ± 2.4 µm, n = 62; p=0.045, Kolmogorov–Smirnov test, mean ± SE) (*Figure 6D*). An explanation that would account for both the enhanced speed and run length is that binding of two Egl molecules favors recruitment of two dimeric dynein motors to the dynactin-BicD-mRNA complex (*Grotjahn et al., 2018*; *Urnavicius et al., 2018*). This result is particularly striking in light of the fact that experiments performed up to this point used a molar ratio of 1 dynein per dynactin to assemble the mRNP, as this was the assumed stoichiometry of binding.

To further investigate whether *K10* mRNA can recruit two molecules of Egl, we mixed the two components in solution, adhered them to a glass coverslip, and examined their color distribution via TIRF microscopy. The numbers reported below reflect correction for complexes with two copies of the same color (see Materials and methods). Controls showed 9.9% colocalilzation of two Egl molecules (Qdot-525 and Qdot-565), and 6.8% colocalization of two *K10* mRNAs (Alexa 488 and Andy 647) (*Figure 6E*). In contrast, when two different colored Egl molecules (Qdot-525 and Qdot-565) were incubated with unlabeled mRNA, 83% of complexes contained two Egl molecules, similar to the 87% of moving complexes described above. Thus, both in the context of a fully reconstituted mRNP, and in a minimal complex containing only Egl and mRNA, *K10* mRNA binds two Egl molecules.

## mRNPs containing two dyneins move faster and longer

Our results with dual-colored Egl molecules strongly suggest that BicD has the potential to recruit two dynein motors. To test this directly, we labeled recombinant dynein containing an N-terminal biotin tag with either Alexa Fluor 647 (red) or Alexa Fluor 488 (green) (*Figure 7A*). *K10* mRNPs were then reconstituted with equimolar red- and green-labeled dyneins, at a molar ratio of 2 dyneins per dynactin. Single-molecule pulldowns in the presence of AMP-PNP revealed that after correction for complexes containing two copies of the same color dynein, 44% of mRNPs contained two dyneins (n = 274 dual-colored) (*Figure 7B*). Parallel analysis of dynein (two colors), complexed with dynactin and truncated BicD2N showed that 46% of complexes contained two motors (n = 114 dual-colored). Finally, we examined the behavior of dynein-dynactin complexes containing Hook1; complexes with Hook3 have been shown to contain predominantly two dyneins (*Grotjahn et al., 2018*; *Urnavicius et al., 2018*). We similarly observed that 50% of the complexes contained two motors (n = 185 dual-colored).

The motile properties of mRNPs containing two colors of dynein were compared with single-colored complexes containing either one or two dyneins. *Figure 7C* shows kymographs of moving mRNPs containing both single- and dual-colored dyneins. The speed histogram of the dual-colored complexes was best fit with a single Gaussian distribution with a speed of 0.63 ± 0.26 µm/s (n = 40) (*Figure 7D*). In contrast, the single-colored complexes were best fit to a double Gaussian distribution with speeds of 0.40 ± 0.14 and 0.74 ± 0.13 µm/s (n = 44), with the faster speed likely corresponding to a population containing two dyneins of the same color (*Figure 7D*). The speed of the

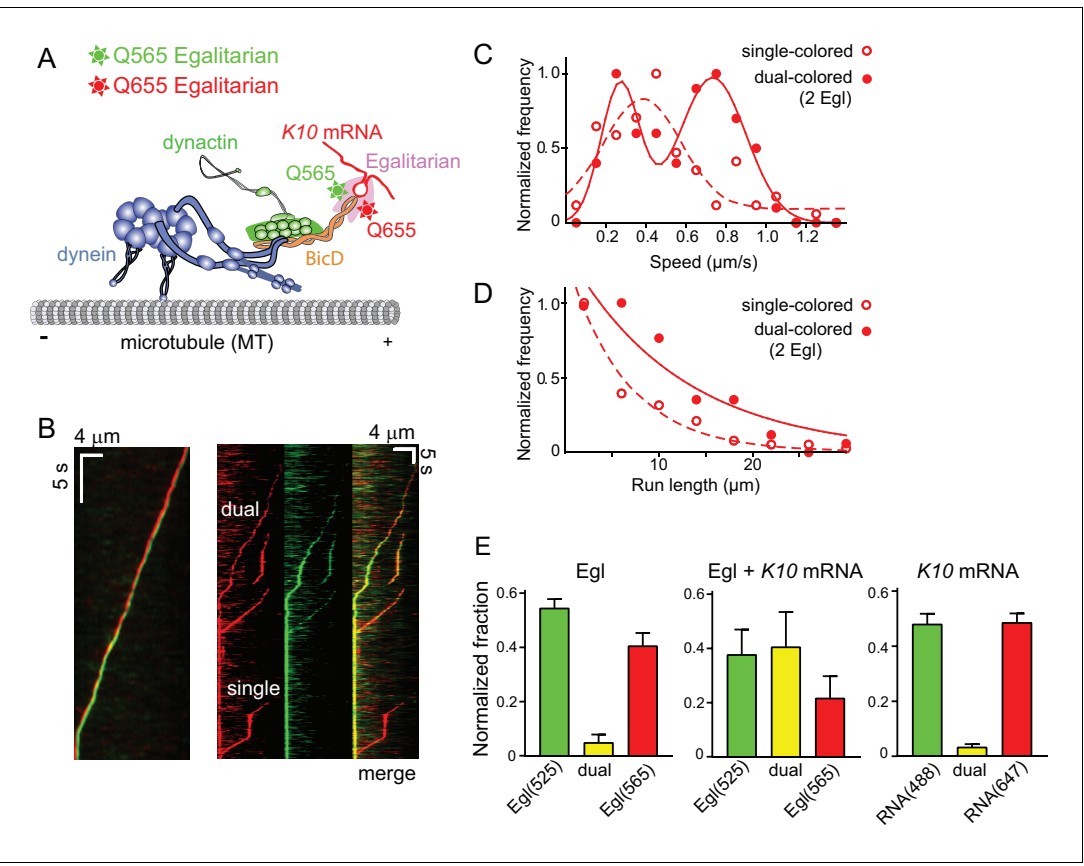

**Figure 6.** Motile properties of complexes containing one versus two Egl molecules. (**A**) Schematic of the two-color experiment in which Egl is labeled with either a 565 or 655 nm QDot (figure adapted from [**Reck-Peterson et al., 2018**]). (**B**) Kymographs of dual-color runs. (Left panel) trajectories of a run with both a 565 nm (green) and a 655 nm (red) Qdot-labeled Egl bound to the moving complex. The red trajectory is shifted horizontally for presentation purposes. (Right panels) Red, green and combined channels of several processive runs. The top three trajectories are dual-color runs, while the bottom one is a single-color run. (**C**) Speeds of single-color complexes (open red circles) were best fit to a single Gaussian distribution with a speed of $0.38 \pm 0.20$ μm/s (n = 81). In contrast, the dual-colored complex (filled red circles) was best fit with a double Gaussian distribution with a slow speed of $0.27 \pm 0.1$ μm/s, and a faster speed of $0.73 \pm 0.16$ μm/s (n = 62). (**D**) Run length histogram of single-color events (open red circles) ($6.4 \pm 0.7$ μm, n = 81) are significantly shorter than dual-color runs (filled red circles) ($12.1 \pm 2.4$ μm, n = 62; p=0.045, Kolmogorov–Smirnov test, mean ± SE of the fit). The curves are exponential fits to the data. (**E**) Dual-colored Egl (525 or 565 Qdots), dual-colored Egl plus mRNA, or dual-colored mRNA (Alexa 488-UTP or Andy 647-UTP) mixed in solution and adhered to a glass coverslip. The percent single- and dual-colored complexes are indicated. After correction for complexes with two copies of the same color Egl (see Materials and methods), 83% of complexes were determined to contain two molecules of Egl. See **Figure 6—source data 1**.
DOI: https://doi.org/10.7554/eLife.36306.020

The following source data is available for figure 6:

**Source data 1.** Dataset for **Figure 6**.
DOI: https://doi.org/10.7554/eLife.36306.021

dual-colored complexes was 58% faster than the single-colored complex slower speed. Run lengths of dual-colored runs were 53% longer than the single-colored complexes ($8.9 \pm 0.5$ μm, n = 40, vs. $5.8 \pm 1.1$ μm, n = 44; p=0.036, Kolmogorov–Smirnov test, mean ± SE) (**Figure 7E**). Recruitment of two dyneins thus results in an mRNP that moves both faster and longer than an mRNP containing only one dynein. **Video 2** shows a clear example of a dual-colored complex that moves faster and longer than single-colored complexes.

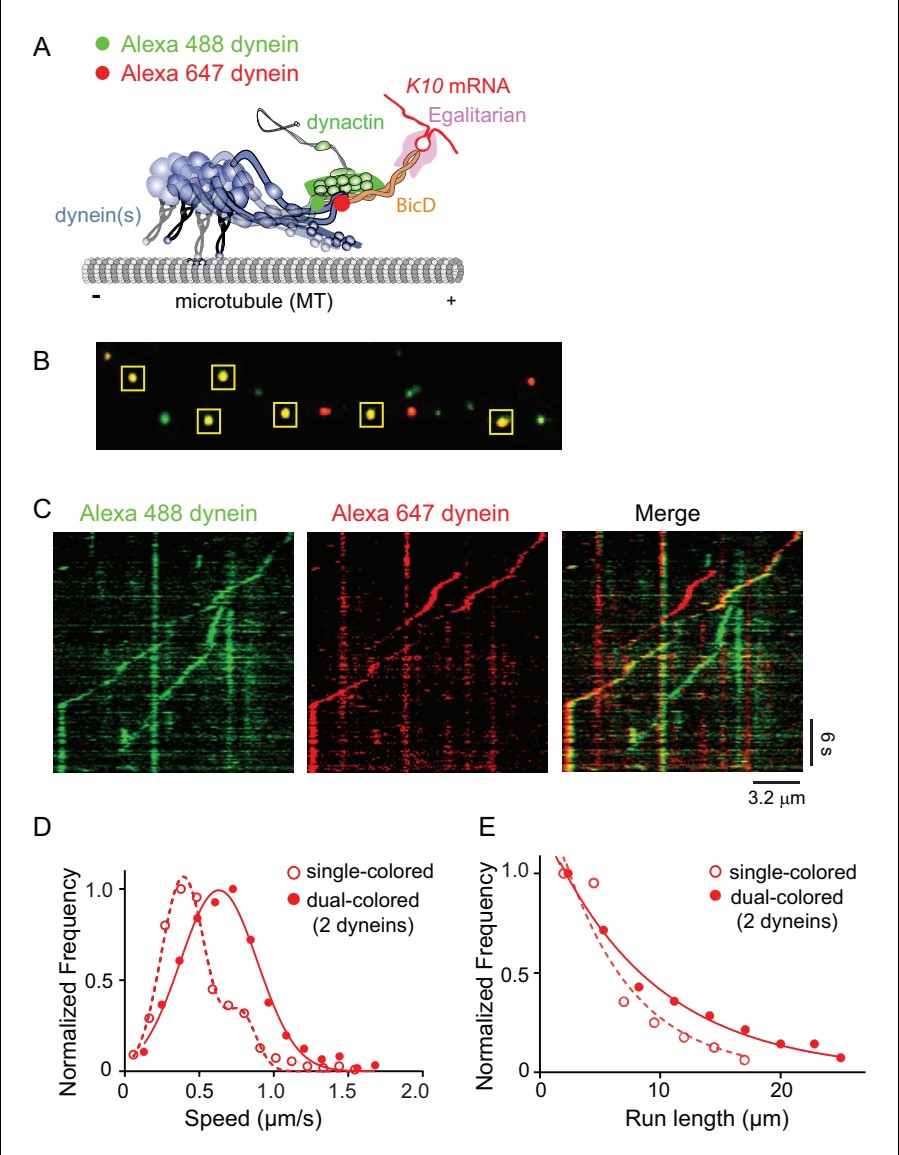

**Figure 7.** Recruitment of two dynein motors to the mRNP results in faster and longer runs. (**A**) Schematic of the two-color experiment. Dynein was either labeled with Alexa 488 (green) or Alexa 647 (red) for the single-molecule pulldowns shown in panel B. mRNPs were formed by incubating with two dyneins per dynactin (figure adapted from [**Reck-Peterson et al., 2018**]). (**B**) Single-molecule pulldowns in the presence of AMP-PNP showed that 22% of complexes were dual-colored, implying that 44% of complexes had two dynein motors bound. (**C**) Kymographs showing single- and dual-colored runs, with dynein labeled with either a 525 or a 655 nm Qdot. Speed and run length are quantified in panels D and E. (**D**) The speed of dual-colored complexes (filled red circles) (0.63 ± 0.26 μm/s, n = 40), were compared to that of single-colored complexes (open red circles) (0.40 ± 0.14 μm/s and 0.74 ± 0.13 μm/s, n = 44). (**E**) Run lengths of dual-colored runs (filled red circles) (8.9 ± 0.5 μm, n = 40) were 53% longer than the single-colored complexes (open red circles) (5.8 ± 1.1 μm, n = 44, p=0.036, Kolmogorov–Smirnov test, mean ± SE of the fit). See *Figure 7—source data 1*.

DOI: https://doi.org/10.7554/eLife.36306.022

The following source data is available for figure 7:

**Source data 1.** Dataset for *Figure 7*.
DOI: https://doi.org/10.7554/eLife.36306.023

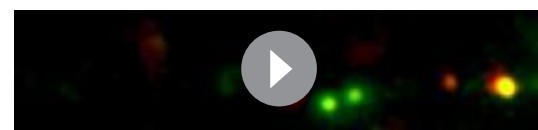

**Video 2.** mRNPs with two dyneins move faster and longer (related to *Figure 7*). The DDBE plus *K10* mRNA complex containing two dimeric dyneins (yellow due to colocalization of a red and green Qdot) moved 12.2 µm in 20.4 s at a speed of 0.6 µm/s on a microtubule track (unlabeled). For comparison, a single dynein (red Qdot) moved a shorter distance (4.5 µm) at a slower speed (0.36 µm/s). The image was magnified twofold and played at 6x real time.

DOI: https://doi.org/10.7554/eLife.36306.024

## The absence of LC8 does not affect speed or run length of the mRNP

Egl can bind the dynein light chain LC8 (DDLC1 in *Drosophila*) through a consensus light chain binding site on Egl at amino acids 963–969 ([963]AESQTLS[969]) (*Navarro et al., 2004*). To determine whether this interaction affects the motility of the mRNP we expressed recombinant dynein with or without LC8 (*Figure 8A*), and reconstituted it into mRNPs at a molar ratio of 2 dyneins per dynactin. As seen with the two-color dynein, the speed distributions were bimodal. We detected no statistical difference in speed between mRNPs reconstituted with WT dynein versus dynein lacking LC8 (0.42 ± 0.13 µm/s and 0.71 ± 0.34 µm/s, n = 396 for WT dynein vs. 0.40 ± 0.13 µm/s and 0.79 ± 0.39 µm/s, n = 488 for dynein without LC8; p=0.57, t-test, mean ±SD) (*Figure 8B*). Likewise, the run lengths showed no statistical difference (6.2 ± 0.11 µm, n = 113 for WT dynein vs. 6.9 ± 0.14 µm, n = 102 for dynein without LC8; p=0.46, Kolmogorov–Smirnov test, mean ± SE) (*Figure 8C*). Thus, the putative interaction between Egl and the dynein LC8 light chain does not change the number of dyneins recruited nor does it change the speed or run length of the mRNP.

## One mRNA per mRNP

Using approaches similar to those described above, *K10* mRNA was labeled with two different colors by incorporating either Andy 488-UTP or Andy 647-UTP (*Figure 9A*). The number of dual-labeled moving mRNPs was 5.5% of the total, the rest being single-colored (*Figure 9B*). After correction for single-colored complexes with two mRNAs, only 11% moving complexes have two mRNAs. Kymographs illustrating this point are shown in *Figure 9C*. Data for speed and run length were similar to experiments performed with unlabeled mRNA. Because the complex was reconstituted with two dyneins per dynactin, the speed pattern was bimodal (0.36 ± 0.09 µm/s and 0.62 ± 0.38 µm/s, n = 187) (*Figure 9D*). Run lengths were 5.5 ± 0.09 µm (n = 67) (*Figure 9E*). The observation that each mRNP contains only one mRNA, along with our previous data showing that 87% of motile mRNPS contained 2 Egls, suggests that a previously unidentified function of the mRNA cargo is to ensure recruitment of two Egl molecules to the complex, favoring disruption of the auto-inhibitory structure of the BicD coiled-coil.

## An Egl-leucine zipper chimera activates motility in the absence of mRNA

To further explore the model that *K10* mRNA ensures recruitment of two Egl molecules to the mRNP complex, we expressed an N-terminal fragment (amino acids 1–121) of Egl containing the BicD binding site that was dimerized by addition of a leucine zipper (Egl[zip]). If this hypothesis is correct, then a motile complex should be formed by dynein-dynactin-BicD-Egl[zip] in the absence of mRNA. Gratifyingly, dynein-dynactin-BicD-Egl[zip] complexes showed motile properties similar to a fully reconstituted mRNP, and were considerably more motile than control complexes composed of dynein-dynactin-BicD-Egl or dynein-dynactin-BicD, as illustrated by kymographs (*Figure 10A*). Although the frequency of movement was somewhat lower than for fully reconstituted mRNPs, it was higher than controls (*Figure 10B*). Speeds of dynein-dynactin-BicD-Egl[zip] were bimodal (blue circles), with the slower population moving at 0.43 ± 0.21 µm/s and the faster population at 0.95 ± 0.15 µm/s (n = 633), whereas speeds of dynein-dynactin-BicD-Egl-*K10* mRNA (red circles) were 0.44 ± 0.31 µm/s (n = 560) (p<0.001, t-test, mean ± SD) (*Figure 10C*). This result is notable for two reasons. First, the fast speed observed with Egl[zip] was higher than previously seen in prior bimodal distributions with native mRNPs reconstituted with two dyneins. Secondly, both complexes were reconstituted with one dynein per dynactin, yet a substantial portion of dynein-dynactin-BicD-Egl[zip] complexes moved at the fast speed indicative of two bound dyneins, solidifying our conclusion

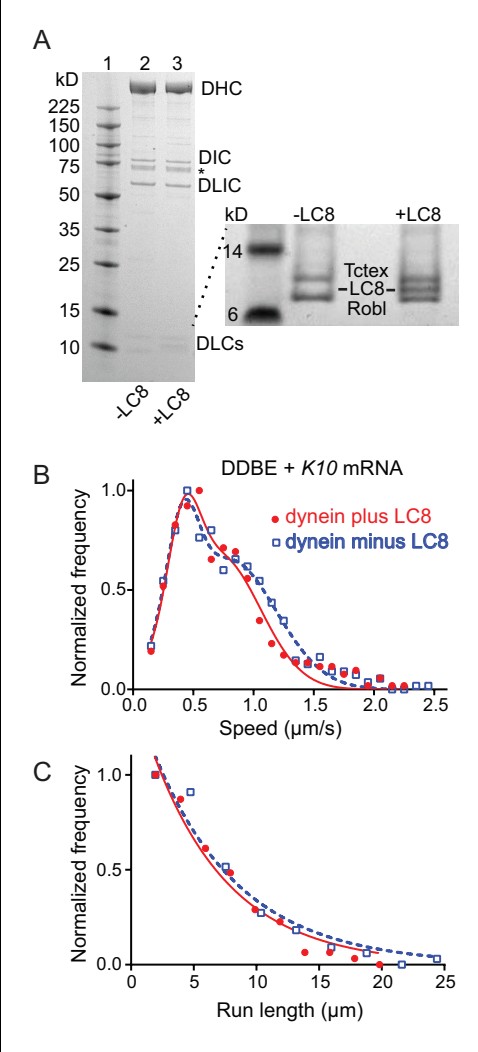

**Figure 8.** Comparison of motile mRNPS reconstituted with WT dynein or dynein without LC8. (**A**) SDS-PAGE of (lane 1) molecular mass markers, (lane 2) expressed WT dynein, (lane 3) dynein expressed without LC8. The identity of the bands labeled DIC and DLIC was confirmed by mass spectrometry. The band marked with an asterisk is a FLAG-reactive fragment that was shown by mass spectrometry to be derived from the heavy chain. DHC, dynein heavy chain; DIC, dynein intermediate chain; DLIC, dynein light intermediate chain. Higher loads of the same samples are shown in the inset so that the light chains can be visualized. 4–12% SDS-PAGE, MES buffer. (**B**) WT dynein speeds (filled red circles) (0.42 ± 0.13 µm/s and 0.71 ± 0.34 µm/s, n = 396) were not statistically different from dynein minus the LC8 light chain (open blue squares) (0.40 ± 0.13 µm/s and 0.79 ± 0.39 µm/s, n = 488; p=0.57, t-test, mean ± SD). (**C**) WT dynein run lengths (6.2 ± 0.11 µm, n = 113) were the same as dynein without LC8 (6.9 ± 0.14 µm, n = 102 r; p=0.46, Kolmogorov–Smirnov test, mean ± SE). Complexes were reconstituted with two dyneins per dynactin. See *Figure 8—source data 1*.

*Figure 8 continued on next page*

(*Figure 6*) that the presence of two Egls favors recruitment of two dyneins. Run lengths of dynein-dynactin-BicD-Egl$^{zip}$ (blue circles) were longer than those of dynein-dynactin-BicD-Egl-*K10* mRNA (red circles) (7.6 ± 0.03 µm, n = 84 vs. 5.7 ± 0.09 µm, n = 106, p=0.03, Kolmogorov–Smirnov test, mean ± SE) (*Figure 9E*). Because runs can terminate due to dissociation of the complex, the presence of the forced Egl dimer may enhance run length by stabilizing the assembled complex. In summary, we conclude that the reason why mRNA is needed for robust mobility of the reconstituted mRNP is to ensure the presence of the two Egl molecules required to relieve BicD auto-inhibition.

## Discussion

Using purified proteins and a synthesized mRNA, we reconstituted a motile mRNP composed of dynein-dynactin, *Drosophila* BicD, the mRNA binding protein Egl, and a localizing mRNA found in *Drosophila* (*K10*). This complex has the capacity to recruit two dimeric dyneins, two Egl molecules, and one *K10* mRNA. Processive motility on microtubules depended on the presence of all components.

In the absence of mRNA, neither full-length BicD nor BicD-Egl bind to dynein-dynactin due to BicD auto-inhibitory interactions. This is in contrast to an N-terminal fragment of mammalian BicD2 (BicD2N) that can recruit dynein-dynactin and convert it into a highly processive motor with enhanced force output, but cannot bind cargo (*Belyy et al., 2016*; *McKenney et al., 2014*; *Schlager et al., 2014*). The comparable truncation of *Drosophila* BicD, BicD$^{CC1}$, also activates dynein-dynactin processive runs in vitro, consistent with studies in *Drosophila* (*Dienstbier et al., 2009*). The speed and run length of both minimal dynein-dynactin-truncated BicD complexes were very similar to those seen with the fully reconstituted mRNP (*Figure 5*).

Class averages of negatively stained full-length BicD revealed the structural basis for BicD auto-inhibition: BicD forms a looped conformation that resembles the letter 'b'. The long straight portion of the letter 'b' corresponds to the first coiled-coil region, CC1 (~40 nm), and the remaining loop to CC2-CC3. An atomic resolution structure of dynactin in complex with BicD2N and the dynein tail showed that 275 amino acids (~40 nm) of BicD bind along dynactin (*Urnavicius et al., 2015*). Although essentially all of the ~40 nm CC1 is exposed in full-length auto-inhibited BicD, binding of CC3 to the middle of CC1 must be

*Figure 8 continued*

DOI: https://doi.org/10.7554/eLife.36306.025

The following source data is available for figure 8:

**Source data 1.** Dataset for *Figure 8*.

DOI: https://doi.org/10.7554/eLife.36306.026

sufficient to block formation of a stable high-affinity ternary complex with dynein-dynactin.

EM images of BicD-Egl retain the same auto-inhibited loop structure seen with BicD alone. Part of the Egl molecule folds into a globular domain which is attached via a flexible linker to the BicD binding site. A candidate for the globular domain is the putative 3′−5′ exonuclease domain identified within Egl (*Mach and Lehmann, 1997*; *Moser et al., 1997*; *Navarro et al., 2004*). Individual residues typically associated with exonuclease activity have been shown to be unnecessary for the function of Egl, whereas deletion of the entire domain strongly reduced RNA-binding activity. The globular domain may represent an RNA binding module that does not require exonuclease activity for normal function.

It is noteworthy that only one globular domain is seen in the EM images, suggesting that at the low protein concentrations used for microscopy and for single-molecule experiments, only one Egl is bound to BicD in the absence of mRNA. Binding studies showed that two Rab6$^{GTP}$ adaptors can bind to the CC3 domain of BicD, by associating with opposite faces of the BicD coiled-coil (*Liu et al., 2013*), but the binding affinity is relatively weak ($K_d$ = 0.9 μM).

## Why is mRNA needed to relieve BicD auto-inhibition?

EM showed that the auto-inhibited loop of BicD was disrupted only in the presence of Egl and mRNA, but not with mRNA alone. This observation is consistent with the requirement of mRNA for high-frequency robust processive motion in single molecule experiments. Experiments using two different colors of the same component, to allow copy number in motile mRNPs to be assessed, lent insight into why mRNA was required. First, 87% of moving mRNPs contained two Egl molecules. Second, two-color mRNA experiments showed that 89% of moving mRNPs contained only one mRNA. Consistent with our in vitro studies, transported mRNAs in *Drosophila* extracts were shown to primarily contain a single mRNA (*Amrute-Nayak and Bullock, 2012*; *Soundararajan and Bullock, 2014*). The observation that two Egls are incorporated into motile mRNPs at the nanomolar concentrations used in single-molecule assays suggested that interactions between components of the assembled complex may enhance affinity, with a likely candidate being the mRNA. If one mRNA tethers two Egl molecules together as our data suggest, Egl essentially becomes bivalent in its interaction with the two-chained coiled-coil BicD, conferring the enhanced avidity associated with bivalent binding (*Siglin et al., 2013*). Thus, two Egl molecules may be required to overcome the auto-inhibition of BicD, and this is enabled by their simultaneous binding to one mRNA. A proof-of-principle experiment that strongly supports this mechanism was the effect of a truncated Egl construct with a leucine zipper, which activates dynein-dynactin-BicD motility in the absence of mRNA to a similar extent as seen for the fully reconstituted mRNP. Tantalizingly, the run lengths and speeds trended to higher values with the zippered Egl than observed with the fully reconstituted mRNP (*Figure 10*), suggesting that this artificial construct may stabilize the motile complex to a greater extent because one Egl can no longer dissociate, which would favor complex disassembly. In vivo, a dynamic complex that needs to disassemble once cargo is transported to its destination may be advantageous and necessary. In a concurrent elegant study, using different *Drosophila* mRNA transcripts and complementary techniques, Bullock and colleagues (*McClintock et al., 2018*) (*see accompanying paper*) also proposed that a single mRNA scaffolds the association of two Egls with BicD CC3 to facilitate activation of dynein. The requirement for mRNA is biologically important because it ensures that motor activation is coupled to cargo binding, preventing futile dynein activity.

We previously showed that in an actomyosin-based mRNA transport system in budding yeast, mRNA was also required for processive motion, but for a different reason: mRNA stabilized the interaction of two single-headed class V myosins (Myo4/She3) with the mRNA-binding adaptor protein She2 at physiological ionic strength (*Sladewski et al., 2013*). Although mechanistically the role of mRNA differs from what we show here with dynein-dynactin and BicD, in both cases the requirement for mRNA ensures that only cargo-bound motor complexes are motile.

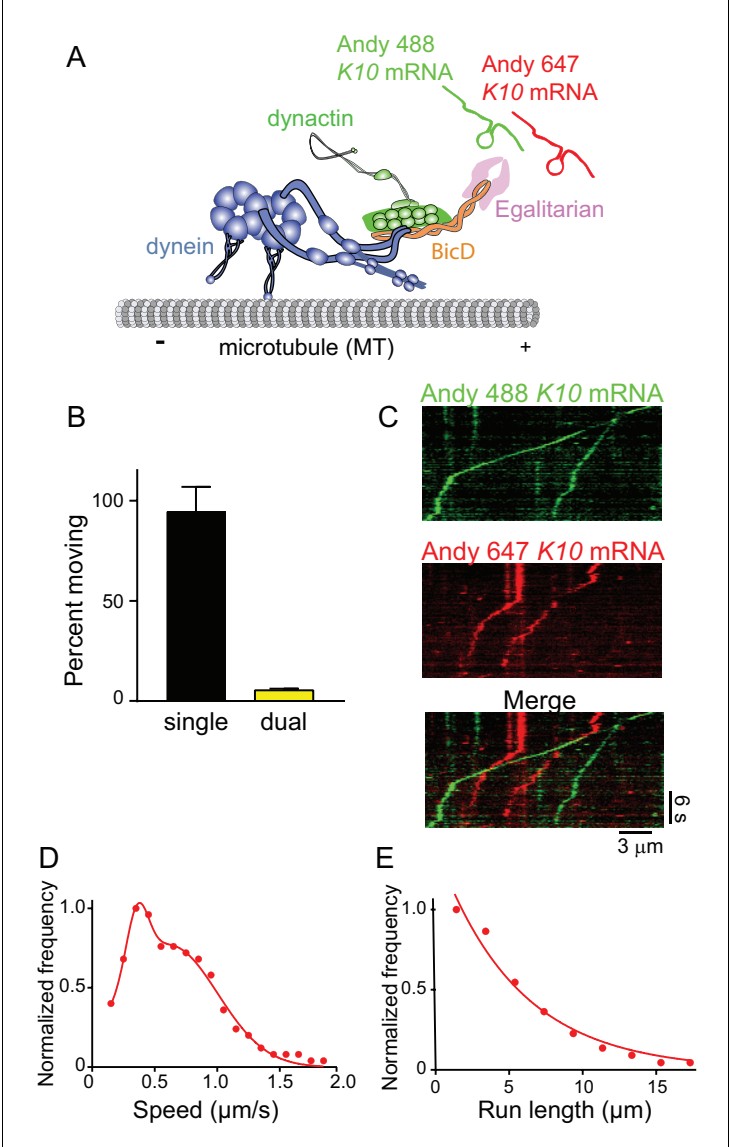

**Figure 9.** The mRNP predominantly contains one *K10* mRNA. (**A**) Schematic of the two-color mRNA experiment. mRNA was either labeled with Andy 488-UTP (green) or Andy 647-UTP (red) (figure adapted from [**Reck-Peterson et al., 2018**]). (**B**) 5.5% of moving complexes were dual-labeled, implying that 11% of the complexes contained 2 mRNAs, and 89% contained one mRNA. (**C**) Representative kymographs highlighting the predominance of single-colored moving complexes. (**D**) The speed pattern was bimodal ($0.36 \pm 0.09$ μm/s and $0.62 \pm 0.38$ μm/s, n = 187) because the complex was reconstituted with two dyneins per dynactin. (**E**) Run lengths were $5.5 \pm 0.09$ μm (n = 67). See *Figure 9—source data 1*.

DOI: https://doi.org/10.7554/eLife.36306.027

The following source data is available for figure 9:

**Source data 1.** Dataset for *Figure 9*.

DOI: https://doi.org/10.7554/eLife.36306.028

## Mechanism by which BicD auto-inhibition is relieved

Steric/competitive or allosteric mechanisms for disruption of the CC3-CC1 interaction by two Egl molecules can be considered. A competitive mechanism suggests that two Egls compete effectively with CC1 for a common binding site on CC3. A competitive mechanism appears to be the case with Rab6$^{GTP}$, an adaptor protein that links dynein-dynactin-BicD to vesicular cargo. The Rab6$^{GTP}$ and CC1 binding sites on CC3 overlap (*Terawaki et al., 2015*), implying that both cannot bind

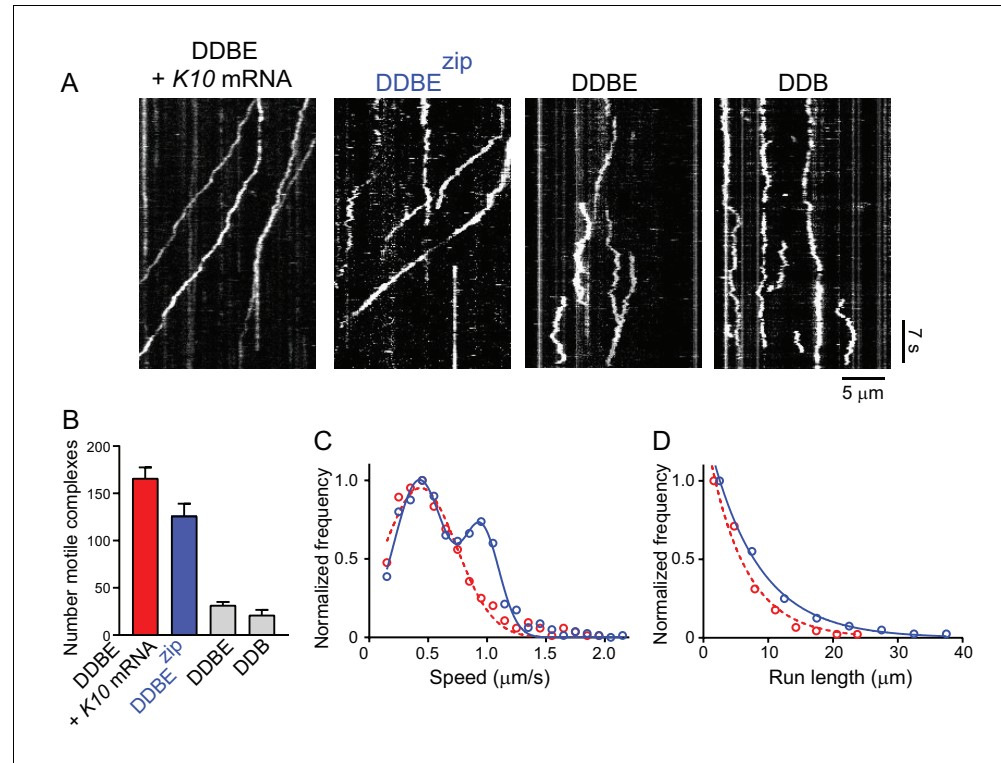

**Figure 10.** A truncated Egl-leucine zipper construct supports motility in the absence of mRNA. (**A**) Kymographs illustrating motion of mRNPS reconstituted from dynein-dynactin-BicD-Egl-*K10* mRNA, dynein-dynactin-BicD-Egl[zip], dynein-dynactin-BicD-Egl, or dynein-dynactin-BicD. (**B**) Run frequency normalized to dynein concentration and time for the same four scenarios illustrated in panel A. (**C**) Bimodal speeds of dynein-dynactin-BicD-Egl[zip] (blue circles), with the slower population moving at 0.43 ± 0.21 μm/s and the faster population at 0.95 ± 0.15 μm/s (n = 633). For comparison, dynein-dynactin-BicD-Egl-*K10* mRNA (red circles) moved at 0.44 ± 0.31 μm/s (n = 560) (p<0.001, t-test, mean ± SD) (**D**) Run lengths of dynein-dynactin-BicD-Egl[zip] (blue circles) were longer than those of dynein-dynactin-BicD-Egl-*K10* mRNA (red circles) (7.6 ± 0.03 μm, n = 84 vs. 5.7 ± 0.09 μm, n = 106; p=0.03, Kolmogorov–Smirnov test, mean ± SE). See *Figure 10—source data 1*.

DOI: https://doi.org/10.7554/eLife.36306.029

The following source data is available for figure 10:

**Source data 1.** Dataset for *Figure 10*.

DOI: https://doi.org/10.7554/eLife.36306.030

simultaneously. Rab6[GTP] (but not Rab6[GDP]) in solution or bound to artificial liposomes released BicD2 from an auto-inhibited state to promote processive dynein-dynactin motion (*Huynh and Vale, 2017*). The bound GTP in essence signals to BicD that the Rab is bound to cargo (reviewed in [*Grosshans et al., 2006*]).

Alternatively, an allosteric mechanism would postulate that binding of Egl-mRNA causes a propagated change in the BicD coiled-coil that weakens the CC3-CC1 interaction (*Liu et al., 2013*). This possibility is due to an unusual feature of part of the *Drosophila* BicD CC3 structure, in which the same residues in the two chains of the coiled-coil adopt different heptad registries, referred to as 'heterotypic' interactions. It was proposed that this region may act as a molecular switch to promote weakening of the CC3-CC1 interaction following cargo binding to the adjacent 'homotypic' segment of coiled-coil (*Liu et al., 2013*).

## Role of the mRNA zip code

The highest frequency of moving complexes, and the highest number of complex associations detected in single-molecule pulldowns, were obtained with *K10* mRNA containing the TLS zip code. These values decreased ~two-fold when the zip code was removed, or with either of two

heterologous mRNA transcripts, but remained higher than in the absence of mRNA, implying that motor complexes may also bind to sequences other than the TLS. Studies using extracts from *Drosophila* embryos also concluded that the role of localization signals/zip codes is to increase the average copy number of dynein-dynactin recruited to an mRNP, but even non-localizing mRNAs associate with dynein and move bidirectionally (*Amrute-Nayak and Bullock, 2012*). The function of zip codes in *Drosophila* appears to differ from that seen in budding yeast, where zip codes in the model mRNA *ASH1* are essential for recruitment of myosin motors to the mRNA-binding adaptor protein She2 (*Sladewski et al., 2013*).

### mRNPs recruit two dyneins

When we began this study, we had assumed that a single dynein was recruited to the complex containing dynein and BicD, and thus used a 1:1 stoichiometry of dynein:dynactin for reconstitutions. Surprisingly, when we reconstituted mRNPs with two different color Egls, we observed that dual-colored complexes had a distinct peak with faster speeds and longer run lengths than single-colored mRNPs containing a mixture of one or two Egls. Separate experiments with dual-colored dyneins directly showed that the enhanced speed and run length can be attributed to the recruitment of two dyneins (*Figure 7*). This result is consistent with recent studies showing that dynactin can recruit two dyneins to a dynein-dynactin-adaptor protein complex (*Grotjahn et al., 2018*; *Urnavicius et al., 2018*), and that complexes with two dyneins show faster movement and enhanced force (*Urnavicius et al., 2018*). In addition, our observation implies a link between a fully activated mRNP and enhanced dynein recruitment. Experiments with a zippered Egl construct, which was also reconstituted with one dynein per dynactin, also showed a distinct faster peak indicative of recruitment of two dyneins, solidifying the link between full occupancy of BicD with two Egls and binding of two dyneins. The putative interaction between Egl and LC8 of dynein does not, however, appear to contribute to dynein recruitment because mRNPs reconstituted with dynein lacking LC8 exhibited the same speed and run length as those containing wild-type dynein.

The cryo-EM studies of *Grotjahn et al. (2018)* visualized two dimeric dyneins bound to essentially all (>97%) of dynein-dynactin-BicD2N or dynein-dynactin-Hook3 complexes bound to a microtubule in the presence of AMP-PNP. Interestingly, Carter and colleagues (*Urnavicius et al., 2018*) also showed that BicDR1 and Hook3 recruited two dimeric dynein motors, but found that only 18% of the BicD2N complexes bound two dyneins, which they related to the position of the N-terminus of the BicD2N fragment in the complex. Our results showed the same number of two-dynein complexes with BicD versus Hook. Factors that influence recruitment of the second dynein to *Drosophila* BicD and human BicD2 remain to be determined. Our studies with full-length *Drosophila* BicD suggest that one factor influencing recruitment of two dimeric dynein motors in an mRNP is the presence of two bound Egls, thus establishing a link between cargo binding and full dynein occupancy on dynactin. This in vitro reconstitution of an mRNP, from motor to bona fide biological cargo, provides an excellent model system to further test interactions between components of the complex, which likely are stabilized by multiple weak interactions that synergize to produce a robust transport complex.

## Materials and methods

**Key resources table**

| Reagent type (species) or resource | Designation | Source or reference | Identifiers | Additional information |
|---|---|---|---|---|
| Chemical compound, drug | Alexa Fluor 488–5-UTP | Molecular Probes | C11403 | |
| Chemical compound, drug | Andy Fluor 488-X-UTP | GeneCopoeia | C410A | |
| Chemical compound, drug | Andy Fluor 647-X-UTP | GeneCopoeia | C418A | |
| Chemical compound, drug | Rnase Inhibitor | Promega | N261B | |

*Continued on next page*

*Continued*

| Reagent type (species) or resource | Designation | Source or reference | Identifiers | Additional information |
|---|---|---|---|---|
| Chemical compound, drug | Q-dot 525 streptavidin conjugate | Invitrogen | Q10141MP | |
| Chemical compound, drug | Q-dot 565 streptavidin conjugate | Invitrogen | Q10131MP | |
| Chemical compound, drug | Q-dot 655 streptavidin conjugate | Invitrogen | Q10121MP | |
| Chemical compound, drug | SNAP-Surface Alexa Fluor 488 | New England BioLabs | S9129S | |
| Chemical compound, drug | SNAP-Surface Alexa Fluor 647 | New England BioLabs | S9136S | |
| Chemical compound, drug | SNAP-Biotin | New England BioLabs | S9110S | |
| Chemical compound, drug | Ribonucleic acid, transfer from *Escherichia coli* | Sigma-Aldrich | R1753 | |
| Chemical compound, drug | Tubulin protein (X-rhodamine): bovine brain | Cytoskeleton, Inc | TL620M-A | |
| Chemical compound, drug | paclitaxel | Cytoskeleton, Inc | TXD01 | |
| Commercial kit | RiboMAX Large Scale RNA Production Systems | Promega | P1280 | |
| Recombinant DNA | pDyn1 (SNAPf-His-ZZ-LTLT-DYNC1H1 in pACEBac1) (*Homo sapiens*) | *Schlager et al. (2014)* | NCBI:NP_001367.2 | Expression plasmids for dynein in *Sf9* cells. See details in Materials and methods. |
| Recombinant DNA | pDyn3 (ZZ-SNAPf-DYNC1H1, DYNC1I2, DYNC1LI2, DYNLT1, DYNLRB1,DYNLL1 in pDynBac1) (*Homo sapiens*) | *Schlager et al. (2014)* | NCBI:NP_001367.2 | Expression plasmids for dynein in *Sf9* cells. See details in Materials and methods. |
| Recombinant DNA | dynein nBiotin tag (*Homo sapiens*) | This paper | NCBI:NP_001367.2 | Expression plasmids for dynein in *Sf9* cells. See details in Materials and methods. |
| Recombinant DNA | Bicaudal D, isoform A (*Drosophila melanogaster*) | This paper | NCBI:NP_724056.1 | Expression plasmids for BicD, YFP-BicD in *Sf9* cells, and BicD$^{CC1}$ in *E. coli*. see details in Materials and methods. |
| Recombinant DNA | Bicaudal D homolog 2 isoform 2 (*Homo sapiens*) | This paper | NCBI:NP_056065.1 | Expression plasmids for BicD2N in *E. coli*. See details in Materials and methods. |
| Recombinant DNA | Egalitarian (*Drosophila melanogaster*) | This paper | NCBI:AAB49975.2 | Expression plasmids for Egl in *Sf9* cells, and Egl-ZIP in *E. coli*. See details in Materials and methods. |
| Recombinant DNA | *K10* mRNA (*Drosophila melanogaster*, f) | This paper | NCBI:NM_058143.3 | Expression plasmids for *K10* mRNA, *K10* no zip mRNA. See details in Materials and methods. |
| Recombinant DNA | β-actin mRNA (*Rattus norvegicus*) | This paper | NCBI:NM_031144.3 | Expression plasmids for b actin mRNA. See details in Materials and methods. |
| Recombinant DNA | *ASH1* mRNA (*Saccharomyces cerevisiae*) | *Sladewski et al. (2013)* | NCBI:NM_001179751.1 | Expression plasmids for Ash1 mRNA. See details in Materials and methods. |
| Recombinant DNA | kinesin G235A (*Mus musculus*) | This paper | NCBI:NM_008449.2 | Expression plasmids for rigor kinesin in *E. coli*. See details in Materials and methods. |
| Biological sample | dynein - dynactin | Bovine brain | | See details in Materials and methods. |

*Continued on next page*

*Continued*

| Reagent type (species) or resource | Designation | Source or reference | Identifiers | Additional information |
|---|---|---|---|---|
| Biological sample | tubulin | Bovine brain | | See details in Materials and methods. |
| Software, algorithm | Nikon ECLIPSE Ti microscope | Nikon | | |
| Software, algorithm | Nikon NIS Elements | Nikon | | |
| Software, algorithm | Andor EMCCD Camera | Andor Technology USA | | |
| Software, algorithm | Prism | GraphPad | v7; RRID:SCR_002798 | |

## DNA constructs

Full-length *Drosophila* BicD (NP_724056.1 or NM_165220.3) was cloned into pACSG2 for production of recombinant virus and expression in Sf9 cells. Where indicated, full-length *Drosophila* BicD constructs contained an N-terminal monomeric YFP and a C-terminal FLAG tag, or an N-terminal FLAG followed by a biotin tag for conjugation to a streptavidin-Qdot (Invitrogen). The biotin tag is a 88 amino acid fragment of the biotin carboxyl carrier protein (*Cronan, 1990*). The FLAG tag facilitated purification.

Truncated *Drosophila* BicD$^{CC1}$ (amino acids 21–380), a derivative of the full-length clone described above, with N-terminal His and biotin tags, was cloned into pET19 for expression in bacteria. Other truncations of the full-length *Drosophila* BicD were CC2 (BicD$^{CC2}$, amino acids L318-Q557), or CC2 and CC3 (BicD$^{CC2-CC3}$, amino acids L318-F782), which were cloned into pACSG2 for production of recombinant virus and expression in Sf9 cells. Both BicD$^{CC2}$ and BicD$^{CC2-CC3}$ had an N-terminal Flag followed by a biotin tag.

Truncated human BicD2N (NM_015250 and NP_056065.1), amino acids 25–398, was cloned into bacterial expression vector pET19 with either an N-terminal HIS and biotin tag, or an N-terminal HIS and monomeric YFP tag. This human BicD2N construct aligns with *Mus musculus* BicD amino acids 25–400.

*Drosophila* Egalitarian (isoform B, NP_726360.3) was cloned into pACSG2 with either a C-terminal FLAG, or C-biotin followed by a HIS tag for production of recombinant virus and expression in Sf9 cells. A truncated, dimerized variant of Egalitarian consisting of residues M1-S121, followed by a linker and the GCN4 leucine zipper (AAL09032.1) to ensure dimerization, and a C-terminal HIS tag was cloned into pET3 for bacterial expression.

The N-terminal 402 amino acids of mouse kinesin (NP_032475 and NM_008449.2) with a G235A point mutation was cloned into pET21b for expression of a rigor kinesin used for attachment of microtubules to the flow cell surface.

Codon-optimized human dynein for expression in Sf9 cells (DYNC1H1 (DHC), DYNC1I2 (DIC), DYNC1LI2 (DLIC), DYNLT1 (Tctex), DYNLRB1 (Robl) and DYNLL1(LC8)) was a generous gift from Simon Bullock (*Schlager et al., 2014*). The heavy chain was modified to contain an N-terminal FLAG tag followed by either a biotin or SNAP tag to enable labeling of the heavy chain. Separate recombinant viruses were produced to express each of the associated subunits (except for Robl and Tctex which were present in the same virus). All subunits were under the polyhedrin promoter except for Robl which was under the p10 promoter.

## Reagents used for protein purification

Reagents used for protein expression and purification include: 4-(2-aminoethyl)benzenesulfonyl fluoride (AEBSF, Fisher BioReagents 30827-99-7), phenylmethylsulfonyl fluoride (PMSF, Sigma-Aldrich P7626), Tosyl-L-lysyl-chloromethane hydrochloride (TLCK, Sigma-Aldrich T7254), leupeptin (Thermo Scientific 78435), benzamidine (Sigma-Aldrich B6506), FLAG affinity resin (Sigma-Aldrich A2220), FLAG peptide (Sigma-Aldrich, F3290) and HIS-Select resin (Sigma-Aldrich P6611), and biotin (Sigma-Aldrich B4639).

## Protein expression and purification

Cytoplasmic dynein and dynactin were purified from bovine brain as described previously (*Bingham et al., 1998*), except that the preparation was scaled down to 1.5 brains (~300 g). Alternatively, dynein was expressed in Sf9 cells as described below. Bovine tubulin was purified from brain tissue as described previously (*Castoldi and Popov, 2003*). Protein concentrations were determined using Bradford reagent with BSA as standard.

Dynein and accessory chains were co-expressed in Sf9 cells for ~72 hr at 27°C, harvested, and re-suspended in 10 mM imidazole, pH 7.0, 0.3 M NaCl, 1 mM EGTA, 5 mM $MgCl_2$, 7% sucrose, 2 mM DTT, 0.5 mM AEBSF, 0.5 mM PMSF, 0.5 mM TLCK, 5 µg/ml leupeptin, and 1.3 mg/ml benzamidine. Cells were lysed by sonication, and centrifuged at 257,000 x g for 40 min. The clarified lysate was added to 4 ml FLAG affinity resin and incubated with mixing for 40 min. Resin was transferred to a column and washed with 200 ml FLAG wash buffer (10 mM imidazole, pH 7.4, 0.2 M NaCl, 1 mM EGTA) and eluted with the same buffer containing 0.1 mg/ml FLAG peptide. Peak fractions were concentrated using a Millipore Amicon Ultra-15 centrifugal filter and dialyzed against 5 mM $NaP_i$, pH 7.2, 0.2 M NaCl, 1 mM DTT, 0.1 µg/ml leupeptin, and 50% glycerol for storage at −20°C.

HIS-tagged Drosophila BicD$^{CC1}$ and human BicD2N constructs were expressed in BL21(DE3) bacterial cells. Cells were induced with 0.7 mM IPTG and grown overnight at room temperature in LB broth containing 0.024 mg/ml biotin. Cells were harvested, pelleted, and re-suspended in HIS lysis buffer (10 mM $NaPO_4$, pH 7.4, 0.3 M NaCl, 0.5% glycerol, 7% sucrose, 7 mM β-ME, 0.5 mM AEBSF, and 5 µg/ml leupeptin). Cells were lysed by sonication, clarified at 33,000 x g for 30 min, and the supernatant bound to 3.5 ml of HIS-Select resin. The resin was washed with wash buffer (10 mM $NaPO_4$, pH 7.4, 0.3 M NaCl) containing 10 mM imidazole, followed by four column volumes of wash buffer containing 30 mM imidazole. Protein was eluted in wash buffer containing 200 mM imidazole, and concentrated using a Millipore Amicon Ultra-15 centrifugal filter. Purified protein was clarified 487,000 x g for 20 min and dialyzed against 10 mM imidazole, pH 7.4, 300 mM NaCl, 1 mM EGTA, 50% glycerol, 1 mM DTT, 0.1 µg/ml leupeptin for storage at −20°C.

Full-length *Drosophila* BicD, BicD$^{CC2}$, and BicD$^{CC2-CC3}$ were expressed in Sf9 cells. For constructs with a biotin tag, the media was supplemented with 0.2 mg/ml biotin. Cells were grown for ~72 hr at 27°C, pelleted, and re-suspended in 10 mM imidazole, pH 7.4, 0.3 M NaCl, 1 mM EGTA, 2 mM DTT, 0.5 mM AEBSF, 0.5 mM PMSF, 0.5 mM TLCK, 5 µg/ml leupeptin, 1.3 mg/ml benzamidine. Cells were lysed by sonication and centrifuged at 257,000 x g for 40 min. The clarified lysate was added to the FLAG affinity resin, and incubated with shaking at 4°C for 40 min. The resin was transferred to a column and washed with 200 ml FLAG wash buffer (10 mM imidazole, pH 7.4, 0.2 M NaCl, 1 mM EGTA) and eluted with FLAG wash buffer containing 0.1 mg/ml of FLAG peptide. Peak fractions were concentrated using a Millipore Amicon Ultra-15 centrifugal filter and dialyzed against storage buffer (10 mM Imidazole, pH 7.4, 200 mM NaCl, 1 mM EGTA, 50% glycerol, 1 mM DTT, 0.1 µg/ml leupeptin) for storage at −20°C. Hook1 was expressed and purified as described for BicD.

Egl containing a C-terminal biotin and FLAG tag, or only a FLAG tag, was expressed in Sf9 cells and purified similarly except that cells were infected for only 48 hr. Lysis and wash steps were done in buffers containing 0.3 M NaCl. The protein was stored at −20°C. in 25 mM imidazole, pH 7.4, 300 mM NaCl, 1 mM EGTA, 1 mM DTT, 0.1 µg/ml leupeptin, 50% glycerol.

For Egl-BicD co-expression in Sf9 cells, Egl contained a C-terminal biotin and HIS tag, and BicD contained N-terminal YFP and FLAG tag, or only an N-terminal FLAG tag. Following infection, cells were grown for ~72 hr at 27°C and then harvested. The pellet was re-suspended in HIS lysis buffer (10 mM $NaPO_4$, pH 7.4, 0.25 M NaCl, 0.5% glycerol, 7% sucrose, 0.1% NP40, 0.5 mM DTT, 0.5 mM AEBSF, 0.5 mM PMSF, 0.5 mM TLCK, 5 µg/ml leupeptin, 1.3 mg/ml benzamidine). The cells were lysed by sonication and the lysate was centrifuged 257,000 x g for 40 min. A 5-ml HIS-Select column was prepared by washing the resin with five column volumes of water, three column volumes of 0.5 M imidazole pH 7.4, 20 column volumes of water, and re-equilibrated in five column volumes of HIS lysis buffer without DTT and protease inhibitors. The high-imidazole wash allows for subsequent use of DTT. The clarified lysate was incubated with resin for 40 min and then washed in a column with ~100 ml of 10 mM imidazole wash buffer (10 mM $NaPO_4$, pH 7.4, 0.25 M NaCl, 10 mM imidazole, 0.5 mM DTT, 0.5% glycerol, 0.1% NP40). Protein was eluted in 10 mM $NaPO_4$, pH 7.4, 0.3 M NaCl, 200 mM Imidazole, 0.5 mM DTT, 0.5% glycerol, 0.1% NP40 and dialyzed overnight against 10 mM imidazole, pH 7.4, 0.25 M NaCl, 0.5% glycerol, 0.1% NP40, 1 mM DTT, 0.1 µg/ml leupeptin.

The dialyzed protein was then purified over a FLAG column to remove excess Egl by incubating with FLAG-affinity resin for 60 min, followed by washing and elution as described for BicD purification. Peak elution fractions were combined, dialyzed against 30 mM HEPES, pH 7.4, 0.25 M NaOAc, 2 mM MgOAc, 1 mM EGTA, 0.1 μg/ml leupeptin, and 2 mM DTT, snap frozen in liquid nitrogen, and stored at −80°C. Some co-expressed preparations were purified only on HIS resin with extensive washing to remove extra BicD. Elution fractions which showed approximately equal band intensities for BicD and for Egl were pooled and treated as described for the two column preparation. For negative stain electron microscopy, protein was dialyzed against 30 mM HEPES pH 7.2, 0.25 M KOAc, 2 mM MgOAc, 1 mM EGTA, 1 mM TCEP, 0.1 μg/ml leupeptin, centrifuged 487,000 x g for 20 min, and drop frozen into liquid nitrogen.

Rigor kinesin (G235A), a mutant that binds to microtubules but does not dissociate in the presence of ATP or support microtubule motility, was cloned into pET21a. Expression was induced with 0.4 mM IPTG overnight at room temperature in *E. coli* Rosetta (DE3) (Novagen) in Terrific Broth Media (Invitrogen 22711–022) containing kanamycin. Cells were harvested, re-suspended in lysis buffer (10 mM Hepes pH 7.5, 10 mM NaCl, 1 mM EGTA, 0.25 mM DTT with 0.5 mM AEBSF, 0.5 mM TLCK, and 5 μg/ml leupeptin), and lysed by sonication. After clarification, the buffer was adjusted to a final concentration of 0.2 M NaCl and loaded onto a HIS-Select column equilibrated with lysis buffer containing 0.2 M NaCl. The column was washed with the same buffer containing 10 mM imidazole, then eluted with lysis buffer containing 0.2 M imidazole. Peak fractions were dialyzed against 50% glycerol, 10 mM imidazole, pH 7.4, 0.2 M NaCl, 1 mM EGTA, 1 mM DTT, and 5 μg/ml leupeptin for storage at −20°C.

## *K10* mRNA constructs and synthesis

The 3'UTR of *Drosophila K10* mRNA (NM_058143.3),105–1165 nucleotides past the stop codon, was cloned after the SP6 promoter in the pSP72 vector (Promega) followed by a poly16A tail and an EcoRV site to allow the vector to be bluntly opened for use as a template for RNA transcription. The 43 nucleotide transport/localization signal (TLS) zip code starts 679 base after the start of the 3'UTR (*Serano and Cohen, 1995*). *K10* mRNA constructs contain 574 bases before and 443 bases after the TLS. For the *K10* no zip construct, the TLS element (CTTGATTGTATTTTTAAATTAATTCTTAAAAAC TACAAATTAA) was removed. A minimal *K10* mRNA construct (*K10*$_{min}$) consists of 195 nucleotides that center the TLS element. A minimal *K10* mRNA construct lacking the zip code (*K10*$_{min}$ no zip) is the same sequence without the TLS and contains an additional 43 bases of 3'UTR sequence immediately following *K10*$_{min}$ so that *K10*$_{min}$ and *K10*$_{min}$ no zip are the same size. The DNA template was bluntly linearized and transcribed using a phage SP6 RNA polymerase (RiboMAX system, Promega). Labeling of the *K10* RNA was achieved by adding a mixture of Alexa Fluor 488 (Molecular Probes, Invitrogen) in a 1:10 molar ratio to unlabeled nucleotides. For experiments with two different color mRNAs, *K10* mRNA was labeled with Andy Fluor 488 UTP or Andy Fluor 647 UTP (GenCopoeia).

For control mRNA experiments, two additional mRNAs were used. The full rat β-actin (*Actb*, NM_031144.3) gene, including 78 bases from the 3'UTR followed by the EcoRV site, was cloned into pSP72. The *ASH1* gene of *Saccharomyces cerevisiae* (NM_001179751.1) beginning with the start codon and ending at 1104 bp, minus the E1 zipcode (618–702 bp) was cloned into pSP72 with a polyA$_{11}$ tail followed by an EcoRV site.

## Flow cell preparation

PEGylated coverslips were made using methods adapted from (*Gestaut et al., 2008*). Glass cover slides (Fisher Scientific 12–545 M) were plasma cleaned for 5 min and transferred to glass Coplin jars containing 1 M KOH and then placed in a sonicating water bath for 20 min. Slides were rinsed thoroughly with nanopure water, then 95% ethanol and dried using a nitrogen stream. Slides were then placed in glass Coplin jars containing 1.73% 2-methoxy(polyethyleneoxy)propyltrimethoxysilane (Gelest, Inc SIM6492.7–25 g) and 0.62% n-butylamine (Acros Organics 109-73-9) in anhydrous toluene (Sigma-Aldrich 244511), prepared with glass pipettes. Coplin jars containing slides were then placed in plastic bags, purged with nitrogen and incubated for 1.5 hr at room temperature. Following this incubation, the slides were dipped successively in two beakers containing anhydrous toluene and dried using a nitrogen stream. The slides were immediately made into flow chambers, placed in

50 ml tubes and stored at −20°C. This procedure produces a PEGylated slide surface that contains small gaps for the purpose of microtubule attachment.

## Single-molecule total internal reflection fluorescence (TIRF) microscopy assays

Bovine tubulin was thawed and centrifuged at 400,000 x g for 5 min at 2°C. Tubulin concentration was determined using Bradford reagent and diluted to 100 µM in ice cold BRB80 (80 mM PIPES, pH 6.9, 1 mM EGTA, and 1 mM $MgCl_2$) and supplemented with 1 mM GTP. For generating labeled microtubules, unlabeled tubulin was mixed with 1 µM rhodamine-labeled tubulin (Cytoskeleton, Denver, CO) for a final labeled/unlabeled ratio of 1:100. The tubulin mixture was polymerized by transferring to 37°C water bath for 20 min and stabilized by adding 10 µM paclitaxel (Cytoskeleton, Denver, CO). Stabilized microtubules were kept at room temperature for experiments performed that day. Microtubules could be stored at 4°C for use in experiments within 3 days.

Labeled or unlabeled microtubules were adhered to PEGylated flow chambers using rigor kinesin for attachment. Rigor kinesin was diluted to 0.2 mg/ml in buffer B (30 mM HEPES, pH 7.4, 25 mM KOAc, 2 mM MgOAc, 1 mM EGTA, 10% glycerol, 10 mM DTT) and added to PEGylated flow chambers for 10 min at room temperature. Flow chambers were then washed three times in buffer A containing 2 mg/ml BSA, 0.5 mg/ml κ-casein and 0.5% pluronic F68. Paclitaxel stabilized microtubules were diluted to a final concentration of 1 µM in buffer B containing 10 µM paclitaxel, and added to flow chambers and incubated for 10 min at room temperature. Flow chambers were washed three times with buffer B containing 10 µM paclitaxel to remove unbound microtubules.

For $DDB^{CC1}$ single-molecule motility, BicD2N containing an N-terminal biotin tag was diluted in buffer B300 (30 mM HEPES pH 7.4, 300 mM KOAc, 2 mM MgOAc, 1 mM EGTA, 0.5% pluronic F68, 20 mM DTT) and centrifuged 400,000 x g for 20 min. Protein concentration was determined using Bradford reagent and diluted to 1 µM in B300. Dynein, dynactin and BicD2N were mixed to a final concentration of 100 nM in buffer Go50 (30 mM HEPES pH 7.4, 50 mM KOAc, 2 mM MgOAc, 1 mM EGTA, 2 mM MgATP, 20 mM DTT, 8 mg/ml BSA, 0.5 mg/ml κ-casein, 0.5% pluronic F68, 10 µM paclitaxel and an oxygen scavenger system (5.8 mg/ml glucose, 0.045 mg/ml catalase, and 0.067 mg/ml glucose oxidase; Sigma-Aldrich-Aldrich)). Streptavidin-conjugated 655 quantum dots (Invitrogen) were added at 200 nM and incubated with proteins on ice for 30 min. Samples were diluted in buffer Go50 to a final dynein concentration of 1 nM and added to microtubule adsorbed flow chambers for imaging.

For imaging complexes where *K10* mRNA is labeled, BicD was diluted in B300 and centrifuged 400,000 x g for 20 min. Egl was diluted in B300 supplemented with 40 mM DTT and incubated on ice for 1 hr before centrifuging 400,000 x g for 20 min. Alternatively, co-expressed BicD-Egl complexes were used. Protein was determined using Bradford reagent. BicD and Egl were combined at 1 µM in B300. 50 nM dynein and dynactin, 100 nM BicD-Egl and 50 nM *K10* mRNA, synthesized with an Alexa Fluor 488 UTP for visualization, was mixed in buffer Go150 (buffer Go50 adjusted to a final concentration of 150 mM KOAc) containing 10 units of RNase Inhibitor (Promega N261B) and 0.25 mg/ml tRNA from *E. coli* (Sigma-Aldrich R1753). The order of mixing is dynein-dynactin, BicD-Egl, RNase Inhibitor, tRNA, *K10* mRNA. The mixture was incubated on ice for 45 min and diluted to a final dynein concentration of 1 nM in Go80 (buffer Go50 adjusted to a final concentration of 80 mM KOAc) before imaging.

For imaging of complexes where the adaptors are labeled, either BicD or Egl containing a biotin tag were used for conjugation to quantum dots for visualization. Mixtures contained 50 nM dynein and dynactin, 100 nM BicD and Egl, 50 nM unlabeled *K10* mRNA and 200 nM Streptavidin-conjugated 655 quantum dots (Invitrogen). Complexes were incubated on ice for 45 min and diluted to a final dynein concentration of 1 nM in Go80 before imaging.

For single-molecule pulldowns on microtubules in the presence of AMP-PNP, BicD fused to an N-terminal YFP tag and Egl containing a C-terminal biotin tag were prepared as described above and pre-mixed at 1 µM in B300. Mixtures containing 50 nM dynein and dynactin, 100 nM BicD-Egl, 50 nM unlabeled *K10* mRNA and 200 nM streptavidin-conjugated 655 quantum dots (Invitrogen) were diluted in Go150 supplemented with 10 units of RNase Inhibitor (Promega N261B) and 0.25 mg/ml tRNA (Sigma-Aldrich R1753). Mixtures were diluted so that the final dynein concentration was 1 nM. YFP was used to visualize YFP-BicD and 655 Qdots were used to visualize Egl on rhodamine-labeled microtubules.

For the dual-color Egl experiment, 1 µM of biotinylated Egl was mixed with either 1 µM streptavidin 565 or 655 nm Qdots for 10 min, blocked with a 10-fold molar excess of biotin, and then combined. The combined labeled Egl was then added to a dynein-dynactin-BicD mixture followed by *K10* mRNA. The final molar ratio of components is dynein:dynactin:BicD:Egl:*K10* = 1:1:1:2:1. Mixtures were diluted in buffer Go50 so that the final concentration of dynein in the assay was 1 nM. The 565 and 655 nm channels were simultaneously recorded and combined later in ImageJ 1.47 v.

For dual-color dynein motility experiments, expressed dynein with an N-terminal SNAP tag on the heavy chain was biotinylated with SNAP-biotin (New England BioLabs, S9110S). 2 µM SNAP-dynein was incubated with 4 µM SNAP-biotin substrate (5 mM sodium phosphate, pH 7.5, 140 mM NaCl, 1 mM DTT) for 30 min at 37°C. Excess reagent was removed by overnight dialysis at 4°C in 30 mM HEPES, pH7.4, 300 mM KOAC, 20 mM DTT, followed by clarification at 350,000 x g for 20 min. Dynein was incubated with either 525 nm or 655 nm streptavidin Qdots (molar ratio of 1:2) for 20 min on ice, then blocked with a 20-fold molar excess of biotin for 10 min. For single-molecule AMP-PNP pulldowns on microtubules to determine percent dual-labeled complexes, 1 µM dynein-biotin was incubated with 2 µM streptavidin Alexa Fluor 488 or Alexa Fluor 647 for 30 min on ice, followed by addition of a 20-fold molar excess of biotin to prevent further binding (Invitrogen). Equimolar amounts of the two different colored dyneins (200 nM total) were incubated on ice with dynactin at a molar ratio of 2:1. Then BicD, Egl and *K10* mRNA were added to the dynein-dynactin complex and incubated another 45 min. The final molar ratio of dynein:dynactin:BicD:Egl:*K10* mRNA in DDBE plus K10 mRNA complex was 2:1:1:2:2. RNase Inhibitor (Promega N261B) and tRNA from *E. coli* (Sigma-Aldrich R1753) were added. A minimal DDB[CC1] complex was formed with similar stoichiometry (2:1:1). To observe movement, the complex was diluted in buffer Go50 to a final dynein concentration of 0.5–1 nM. Motion was observed on rhodamine-labeled microtubules using TIRF microscopy and images of the Qdots (525 and 655 nm) were recorded simultaneously. For single-molecule pulldowns to show dynein stoichiometry, dynein was observed on unlabeled microtubules in the presence of 6 mM AMP-PNP.

For the two-color mRNA experiment, 1 µM of tissue-purified dynein was mixed with dynactin at a molar ratio of 2:1, followed by BicD and Egl. An equimolar mixture of Andy Fluor 488 and Andy Fluor 647 labeled *K10* mRNA was then added to the complex. Mixtures were incubated for 45 min. The final molar ratio of components was dynein:dynactin:BicD:Egl:*K10* =2:1:1:2:2. Mixtures were diluted in buffer Go50 so that the final concentration of dynein in the assay was 1 nM. The Andy 488 nm and Andy 647 labeled *K10* mRNA were recorded simultaneously using two separate channels (488 and 641 nm laser lines) and combined later in Image J 1.47 v.

For the zippered Egl experiment, 200 nM of tissue-purified dynein was mixed with dynactin at a molar ratio of 1:1, followed by BicD-Bio and Egl[zipper]. The complex was incubated for 45 min on ice. Streptavidin-conjugated 655 quantum dots (Invitrogen) were added at 400 nM and incubated with proteins on ice for 30 min. The same conditions and molar ratios were applied for the controls.

## Imaging and data analysis

Imaging was carried out on a Nikon ECLIPSE Ti microscope, run by the Nikon NIS Elements software package, and equipped with through-objective type TIRF. The samples were excited with the TIRF field of 405/488/561/640 nm laser lines, and emission was split by an image splitter (561 or 638 nm dichroic) and recorded on two Andor EMCCD cameras (Andor Technology USA, South Windsor, CT) simultaneously at five frames/s with automatic focus correction. The final resolution is 0.1066 µm/pixel. Motile mRNPs were tracked with labeled mRNA or with adaptors labeled with a Qdot, and run lengths were measured with ImageJ and the particle-tracking plug-in MTrackJ (*Meijering et al., 2012*). For all processivity assays, frequencies were generated by counting the total number of runs in movies acquired no more than 15 min after dilution of the mRNP mixture. The total number of runs was divided by the total microtubule length (with a constant time and final dynein concentration for all samples being compared) to generate a run frequency. Speeds were measured by tracking run trajectories every 0.2 s with ImageJ using the particle-tracking plug-in SpotTracker (*Sage et al., 2005*).

## Calculation for stoichiometry of complexes using two different color subunits

We performed experiments using two different colored Egl, two different colored dynein, or two different colored mRNA to determine what percent of complexes contain one versus two copies of these proteins or mRNA. In each case, dual-colored complexes represent two copies of the protein (or mRNA) in the complex, while single-colored complexes represent a mixture of one and two-copies in the complex. From quantifying the number of single- and dual-colored complexes, the following equations from *Urnavicius et al. (2018)* define the fraction of complexes with one or two copies.

Color $1_{obs} = (s \times r) + (d \times r^2)$

Color $2_{obs} = (s \times g) + (d \times g^2)$

Dual-colored$_{obs} = d \times 2(r \times g)$

$r$ is the fraction of color one labeled subunit and $g$ is the fraction of color two labeled subunit used to form the complex, with r + g = 1. Solving for $d$ gives the corrected fraction of complexes that contain two copies, and $s$ is the percentage of single-copy complexes.

## Negative stain electron microscopy and image processing

YFP-BicD, YFP-BicD-Egl, BicD-Egl (expressed and purified as intact complexes) were imaged by diluting to 10–25 nM in buffer containing 30 mM HEPES, pH 7.2, 250 mM KOAc, 2 mM MgOAc,1 mM EGTA, 1 mM TCEP. For experiments with mRNA, YFP-BicD-Egl complex was mixed with a two-fold molar excess of mRNA prior to dilution. A 3 µl volume of diluted samples were applied to UV-treated, carbon-coated copper grids and stained with 1% uranyl acetate. Micrographs were recorded using an AMT XR-60 CCD camera at room temperature on a JEOL 1200EX II microscope at a nominal magnification of 40,000. Catalase crystals were used as a size calibration standard. 2D image processing was performed using SPIDER software as described previously (*Burgess et al., 2004*). Alignments are reference free and based on the SPIDER operations AP RA and AP SA. Classifications are K-means and based on the SPIDER operation CL KM. Data used in this study consisted of the following numbers of images: YFP-BicD = 4205, YFP-BicD-Egl=4117, YFP-BicD-Egl-mRNA=3345, YFP-BicD-mRNA=1393, BicD-Egl = 3697, BicD-Egl-mRNA=3526, mRNA = 2039. To compare YFP-BicD with YFP-BicD-Egl complex, 4205 images of YFP-BicD alone and 4117 images of YFP-BicD-Egl were combined into a single stack and subjected to reference-free alignment. Aligned images were classified into 20 classes using K-means classification and classes in the most common orientation were selected. A substack containing only these particles was generated. This substack (4955 images) was realigned and the BicD only images (n = 2471) and BicD-Egl images (n = 2484) were averaged (*Figure 3G* – Global). The aligned substack was classified into 10 classes using a mask around the loop. Difference images shown in *Figure 3G* and *Figure 3—figure supplement 4F* are the result of subtracting the BicD averages from the BicD-Egl averages. The heatmap shown in *Figure 3H* was created by marking the position of the Egl in 342 images of the aligned BicD-Egl stack described above. To generate the low-resolution 3D map shown in *Video 1*, 200 2D class averages of BicD were aligned to a starting model consisting of a second-order Gaussian ellipsoid. The resulting model was then refined against 4205 raw images of BicD. The median filtered volume and PDB files (PDB ID: 2Y0G, 5AFU, 1YT3) were arranged manually using UCSF Chimera to create the final movie.

## Abbreviations

BicD, full-length *Drosophila* Bicaudal D; BicD$^{CC1}$ coiled-coil 1 domain of *Drosophila* Bicaudal D; BicD2N, coiled-coil 1 domain of mammalian Bicaudal D2; DDB$^{CC1}$, dynein-dynactin-BicD$^{CC1}$; DD (BicdD2N), dynein-dynactin-BicD2N; DDB, dynein-dynactin-BicD complex; DDBE, dynein-dynactin-BicD-Egl complex; Egl, the mRNA-binding protein Egalitarian; Egl$^{zip}$, a truncated Egl construct followed by a leucine zipper; $K10_{min}$,195 nucleotides of *K10* that center the TLS element; Qdot, quantum dot; TLS, transport/localization sequence.

## Acknowledgements

This work was funded by National Institutes of Health grant GM078097 to KMT, and American Heart Association Grant 17GRNT33661158 to MYA. We thank the NHLBI Electron Microscopy Core Facility

for the use of their microscopes. Mass spectrometry analysis (*Figure 8*) was supported by an Institutional Development Award (IDeA) from NIGMS (P20GM103449), and was performed in the VGN Bioinformatics and Microarray Facility at the University of Vermont. We thank Mark McClintock and Simon Bullock for sharing their unpublished results.

## Additional information

### Funding

| Funder | Grant reference number | Author |
| --- | --- | --- |
| National Institutes of Health | GM078097 | Kathleen M Trybus |
| American Heart Association | 17GRNT33661158 | M Yusuf Ali |

The funders had no role in study design, data collection and interpretation, or the decision to submit the work for publication.

### Author contributions
Thomas E Sladewski, Neil Billington, M Yusuf Ali, Hailong Lu, Conceptualization, Data curation, Formal analysis, Investigation, Methodology, Writing—original draft, Writing—review and editing; Carol S Bookwalter, Resources, Data curation, Investigation, Methodology, Writing—original draft, Writing—review and editing; Elena B Krementsova, Resources, Investigation, Methodology; Trina A Schroer, Resources, Writing—review and editing; Kathleen M Trybus, Conceptualization, Supervision, Funding acquisition, Methodology, Writing—original draft, Project administration, Writing—review and editing

### Author ORCIDs
Neil Billington http://orcid.org/0000-0003-2306-0228
Trina A Schroer http://orcid.org/0000-0002-5065-1835
Kathleen M Trybus http://orcid.org/0000-0002-5583-8500

### Decision letter and Author response
Decision letter https://doi.org/10.7554/eLife.36306.033
Author response https://doi.org/10.7554/eLife.36306.034

## Additional files

### Supplementary files
• Transparent reporting form
DOI: https://doi.org/10.7554/eLife.36306.031

### Data availability
Data generated or analyzed during this study are included in the manuscript and supporting files. Source data files have been included for Figures 1, 4-10.

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
