## [Decision Letter]

Thank you for submitting your article "Recruitment of two dyneins to an mRNA-dependent Bicaudal D transport complex" for consideration by *eLife*. Your article has been reviewed by Andrea Musacchio as the Senior Editor, a Reviewing Editor, and three reviewers. The reviewers have opted to remain anonymous.

The reviewers have discussed the reviews with one another and the Reviewing Editor has drafted this decision to help you prepare a revised submission.

Summary:

Sladewski and colleagues present the first reconstitution of dynein-mediated RNA transport using purified components. Using single-molecule assays they show that robust transport requires the presence of full-length BicD, mRNA and the mRNA-binding protein Egalitarian (Egl). They also show that the presence of two Egl leads to faster and more processive movement and that faster, more processive events are associated with the recruitment of a second dynein molecule to the complex. The authors also used negative stain electron microscopy to characterize the architecture of the autoinhibited conformation of BicD, and to show that both Egl and the mRNA are required to open up BicD to its active form.

The reviewers were positive about the work but also identified weaknesses that require your attention, and that leave the overall impression that the results are interesting but still somewhat preliminary. In particular, reviewer #1 notes that there is no mechanistic insight into the role of mRNA in RNP assembly. In addition, there are concerns with technical aspects of the work, in particular the validity of comparing a human BICD2 fragment (CC1) with full-length *Drosophila* BicD (Figure 1 and Figure 5) and the lack of some important controls). Reviewer #3 pointed out that the very minimal data processing done with the EM data and the fact that the stoichiometry of the RNA in the transport complex was not addressed directly experimentally are major weaknesses of the current version of the manuscript. The current EM analysis is well below what would be considered acceptable in the field.

These comments were discussed among reviewers and there was an agreement that the points below should be dealt with as clearly and thoroughly as possible in a revised version of the manuscript.

Essential revisions:

1) In Figure 1, protein gels of the dynein, dynactin, and BICD2 CC1 preparations need to be shown. It is not clear why the Egl preparation is shown in this figure, as Egl is not introduced until Figure 3.

2) In Figure 1, the authors should clearly state that dynein and dynactin come from bovine brain, the CC1 fragment of BICD2 is human, and the full-length BicD protein is from *Drosophila*. If it is not possible to repeat experiments with *Drosophila* BicD CC1 within the required time frame, the caveat of species differences should at least be clearly acknowledged. Point 3 below addresses the same concern.

3) In Figure 1, in the pull-down assay (Figure 1C, D), the authors observe that BICD2 CC1 "showed 13-fold enhanced recruitment to dynein-dynactin bound to microtubules compared with full-length BicD". Is that due to autoinhibition of BicD, as is implied in the text, or simply because BICD2 CC1 is from another vertebrate species, whereas BicD is from insects? A more valid comparison would be between *Drosophila* full-length BicD and *Drosophila* BicD CC1.

4) In Figure 1, the controls for unspecific binding of BICD2 CC1 and BicD to microtubules should be shown.

5) In Figure 1, what do the error bars in Figure 1D represent?

6) In Figure 2, what is the difference between "d" and "b" shapes? The EM images in rows 1 and 2 of Figure 2A look the same.

7) In the context of the discussion of Figure 2, the authors mention pull downs were performed with CC1 vs CC2-CC3. These should be shown.

8) The protein gel of the Egl preparation belongs to Figure 3.

9) In Figure 3, a conformational change in the YFP:BicD-Egl complex is observed when mRNA is added. An important missing control is to add mRNA to BicD alone to show that Egl is required.

10) In Figure 4, there is a 2-fold drop in recruitment to microtubules between RNPs containing *K10* mRNA and *K10* mRNA without the transport/localization sequence (TLS). The authors speculate that regions in *K10* mRNA other than the TLS could contribute to Egl binding. Another possibility is that any mRNA can un-specifically interact with Egl in this assay. Although the authors mention that tRNAs do not activate DDBE complexes, a better control would be to use another mRNA or a mix of mRNAs.

11) In Figure 4D: shifting the two channels relative to each other in the merged image is not appropriate; better to show the channels in separate.

12) LC8 data:

- why not present the results in graphs as done for Figures 5 – 7?

- the authors state that they purified dynein without LC8. The protein gel of this preparation should be shown.

13) In Figure 5, as in Figure 1C,D, human BICD2 CC1 is compared with *Drosophila* full-length BicD, which is not ideal.

14) Figure 2A. Did the authors try to mirror the particles in (2) and see if that leads to particles/averages like those in (1)? It seems as though the particles in (2) fell in the flipped orientation relative to those in (1). The concern is that calling them "a less common "d" configuration" implies structural (conformational) differences that may not exist.

15) Did the authors attempt any classification of their data? It was surprising that only single particles and global averages are shown. Even though the data sets are of relatively modest size (~4,000 particles or less), the contrast in the single particles is very good, which means that it should be possible to get improved signal-to-noise even with averages containing a small number of particles.

16) What are the variances in Figure 2C supposed to be telling us? If anything, the fact that the regions of high variance mirror the global average shown on the left tells us that the data is a mixture that has been aligned to the predominant view and that classification is therefore required.

17) Where is the rest of Egl in Figure 3A? Some classification (maybe targeted to different regions of BicD) should reveal additional densities when compared with the BicD alone data. The authors should perform difference mapping between class averages of BicD and BicD-Egl; just one difference map would suffice.

18) Did the authors attempt to label the mRNA in Figure 3D (with gold, for example)? How do we know the mRNA is present in these particles? Depending on the data, this is something that might also be addressable with classification.

19) In Figure 3C, it might be more informative to subclassify b-type data using masks around the b-loop. That should reveal different positions of Egl and those averages could be shown instead of the heat map. Similarly, the difference map shown in Figure 3C is confusing; the fact that there are strong differences surrounding the b-type average, and that these differences do not match the heat map, suggests that the difference map is mostly reporting on the differences between the b-type averages compared. A comparison of class averages (rather than global averages) would reveal whether these differences are relevant (i.e. the loop changes in the presence of Egl) or simply report on the general variability among particles.

20) What is the stoichiometry of the mRNA in the fast, highly processive transport complexes? This seems like an important aspect of the model that was not addressed experimentally. The authors could perform an experiment analogous to those done with dynein and Egl with mRNAs labeled with two different dyes.

21) The cartoons of the transport complexes are disappointing as they reflect so little of our structural understanding of these complexes. Conveying realistic spatial relationships is important for broader audiences who are less familiar with these complex machines. A few of the issues are: (1) Dynactin is shown as a rounded rectangle that represents neither its structure nor its size relative to dynein. This is particularly surprising when one of the authors of the paper is the person who discovered dynactin and was involved in two nice papers on the structure of DDB complexes. (2) Dynein is shown with 2-fold symmetry, which we now know is precisely what dynactin breaks to impose translational symmetry and activate the motor. (3) Coiled-coils are represented as globular domains, while they should look more like dynein's stalk (the only coiled-coil shown as such) or p150. (4) CC1 of BicD should be sandwiched between dynein and dynactin, reflecting its activating role; the figure shows it interacting only with dynactin.

---

## [Author Response]

Summary:Sladewski and colleagues present the first reconstitution of dynein-mediated RNA transport using purified components. Using single-molecule assays they show that robust transport requires the presence of full-length BicD, mRNA and the mRNA-binding protein Egalitarian (Egl). They also show that the presence of two Egl leads to faster and more processive movement and that faster, more processive events are associated with the recruitment of a second dynein molecule to the complex. The authors also used negative stain electron microscopy to characterize the architecture of the autoinhibited conformation of BicD, and to show that both Egl and the mRNA are required to open up BicD to its active form.The reviewers were positive about the work but also identified weaknesses that require your attention, and that leave the overall impression that the results are interesting but still somewhat preliminary. In particular, reviewer #1 notes that there is no mechanistic insight into the role of mRNA in RNP assembly. In addition, there are concerns with technical aspects of the work, in particular the validity of comparing a human BICD2 fragment (CC1) with full-length Drosophila BicD (Figure 1 and Figure 5) and the lack of some important controls). Reviewer #3 pointed out that the very minimal data processing done with the EM data and the fact that the stoichiometry of the RNA in the transport complex was not addressed directly experimentally are major weaknesses of the current version of the manuscript. The current EM analysis is well below what would be considered acceptable in the field.

We provide a substantially revised manuscript that we believe addresses the major weaknesses as described above. New experiments were done to show that one RNA binds per mRNP, and that a zippered Egl will activate the complex in the absence of mRNA. Taken together, these new data provide the mechanistic insight that two Egls are required to overcome the auto-inhibition of full-length BicD, which is orchestrated by the mRNA cargo vis à vis its role in coupling two Egl molecules together. As requested, we also expressed a truncated *Drosophila* BicD (BicD^CC1^) and characterized it in single molecule pulldowns (Figure 1) and in moving complexes (speed, run length) (Figure 5). Not surprisingly, it showed the same properties as mammalian BicD2N, confirming that the different properties of truncated and full-length BicD are not due to species differences. Concerns about the EM have been addressed by revision of Figure 2 and Figure 3, and inclusion of 5 additional supplemental figures to specifically address the reviewer’s comments.

These comments were discussed among reviewers and there was an agreement that the points below should be dealt with as clearly and thoroughly as possible in a revised version of the manuscript.Essential revisions:

For ease of reading, we grouped and responded to all of the comments related to the electron microscopy analysis first (points 6, 9 and 14-19).

6) In Figure 2, what is the difference between "d" and "b" shapes? The EM images in rows 1 and 2 of Figure 2A look the same.

The only difference between the “d” and “b” shapes is likely to be the particle orientation on the EM grid, as suggested by the reviewers. This issue is raised multiple times throughout the reviewer comments and it is clear that we did a poor job of explaining this clearly in the manuscript.

The particle has a preferred orientation on the grid which leads to the dominant appearance resembling that of a letter “b”. A less common appearance, which is almost certainly due to the same conformation being in the opposite orientation on the grid, results in the appearance being that of a letter “d”. In addition, there are occasional but rarer side views.

The purpose of showing these two orientations in Figure 2 was firstly to introduce the most common, “b type”, orientation. Since this orientation dominates, it is used in subsequent analysis to simplify the process of locating variable portions of the images. By analyzing only the molecules in which the loop is in a relatively fixed position, variability in the position of other parts of the molecule, relative to the loop, can be assessed. Secondly, as pointed out by the reviewers, the b and d type image look simply like mirror images of each other. If the precise orientation was strongly affected by, for example, a large domain on one face of the loop, then the molecule would be less likely to lie flat when that face was presented to the carbon. This suggests that in these orientations, the molecule is quite flat in the plane of the carbon. We believe that this is best appreciated by showing examples of both orientations.

Since the confusion stems from our poor wording, we have amended the text. We now make clear that we interpret these appearances to be different orientations of the same conformation. All uses of the word configuration have been removed.

9) In Figure 3, a conformational change in the YFP:BicD-Egl complex is observed when mRNA is added. An important missing control is to add mRNA to BicD alone to show that Egl is required.

The data for the mRNA + BicD control have now been added. We did not observe a disruption of the loop in the presence of mRNA alone, suggesting that only the combination of mRNA and Egl is sufficient to destabilize the autoinhibited conformation.

14) Figure 2A. Did the authors try to mirror the particles in (2) and see if that leads to particles/averages like those in (1)? It seems as though the particles in (2) fell in the flipped orientation relative to those in (1). The concern is that calling them "a less common "d" configuration" implies structural (conformational) differences that may not exist.

As discussed regarding point 6 above, we do not interpret them to be a different conformation, we interpret them to be a different orientation, as suggested by the reviewer. The wording has been changed to make this clear.

15) Did the authors attempt any classification of their data? It was surprising that only single particles and global averages are shown. Even though the data sets are of relatively modest size (~4,000 particles or less), the contrast in the single particles is very good, which means that it should be possible to get improved signal-to-noise even with averages containing a small number of particles.

We did and had elected to omit those from the manuscript for the sake of brevity. Since it is clear that the classifications can be useful for readers in interpreting the data, we have now included classifications in the supplementary data. Example class averages are also included in the main figures. The results of the classifications strongly depends on the masks used, due to variable positions of multiple components. We have included classification using masks focused on different regions for this reason.

16) What are the variances in Figure 2C supposed to be telling us? If anything, the fact that the regions of high variance mirror the global average shown on the left tells us that the data is a mixture that has been aligned to the predominant view and that classification is therefore required.

The intention of the variances in Figure 2C was to illustrate the variable position of the region of CC1 which extends beyond the loop. As pointed out by the reviewer this is made less obvious by the regions of high variance around the loop itself. The classifications requested by the reviewers are now included as supplementary material. The global variance image has been moved to the supplementary material as has the variance image following the second round of alignment (Figure 2—figure supplement 1).

17) Where is the rest of Egl in Figure 3A? Some classification (maybe targeted to different regions of BicD) should reveal additional densities when compared with the BicD alone data. The authors should perform difference mapping between class averages of BicD and BicD-Egl; just one difference map would suffice.

The classifications are now included. Using a mask around the region adjacent to the loop reveals the globular domain attributed to the attached Egl. A concern with this analysis was the presence of the YFP-CC1 region which in a minority of cases can occupy the region adjacent to the loop. In order to mitigate these concerns, we also aligned and classified images of BicD-Egl lacking the YFPs. The results were very similar, with the globular domain still present next to the loop.

Although we had done difference mapping of class averages, the results were quite noisy and we had thus elected not to include them. Examples of these are now included as part of Figure 3G, with the full classification in the supplement. The additional density is seen as a region centrally and towards the bottom of the loop.

18) Did the authors attempt to label the mRNA in Figure 3D (with gold, for example)? How do we know the mRNA is present in these particles? Depending on the data, this is something that might also be addressable with classification.

We were not able to successfully label the mRNA in such a way as to allow the various components of the activated complex to be identified. We believe that the marked difference between YFP-BicD-Egl and YFP-BicD-Egl-mRNA is sufficient to demonstrate the presence and effect of the mRNA. This can be seen by comparing fields of view of the different datasets, individual raw images of particles and also the classifications of the data. We have now included all of these to make this difference clearer to readers. Alignment of the complex in the presence of RNA is largely unsuccessful due to the variable appearance of the complex. In contrast, the presence of mRNA alone (i.e. without Egl), is insufficient to cause a disruption of the loop. The disruption of the loop is also observed when we compare BicD-Egl with BicD-Egl-mRNA (version lacking YFP) and the data showing this are now included.

19) In Figure 3C, it might be more informative to subclassify b-type data using masks around the b-loop. That should reveal different positions of Egl and those averages could be shown instead of the heat map. Similarly, the difference map shown in Figure 3C is confusing; the fact that there are strong differences surrounding the b-type average, and that these differences do not match the heat map, suggests that the difference map is mostly reporting on the differences between the b-type averages compared. A comparison of class averages (rather than global averages) would reveal whether these differences are relevant (i.e. the loop changes in the presence of Egl) or simply report on the general variability among particles.

These classifications are now included.

Responses to non-EM questions follow:

1) In Figure 1, protein gels of the dynein, dynactin, and BICD2 CC1 preparations need to be shown. It is not clear why the Egl preparation is shown in this figure, as Egl is not introduced until Figure 3.

SDS-PAGE gels of expressed full-length BicD, *Drosophila* BicD ^CC1^ and human BicD2Nand tissue purified dynein and dynactin are now in Figure 1—figure supplement 1. SDS-gels of expressed Egl preparations are now in Figure 3—figure supplement 1.

2) In Figure 1, the authors should clearly state that dynein and dynactin come from bovine brain, the CC1 fragment of BICD2 is human, and the full-length BicD protein is from Drosophila. If it is not possible to repeat experiments with Drosophila BicD CC1 within the required time frame, the caveat of species differences should at least be clearly acknowledged. Point 3 below addresses the same concern.

We chose to express a truncated *Drosophila* BicD^CC1^ and characterized it in single molecule pull downs (Figure 1) and in moving complexes (Figure 5). We saw no differences between *Drosophila* BicD^CC1^ and mammalian BicD2N, and thus our conclusions stand and are not due to species differences.

3) In Figure 1, in the pull-down assay (Figure 1C, D), the authors observe that BICD2 CC1 "showed 13-fold enhanced recruitment to dynein-dynactin bound to microtubules compared with full-length BicD". Is that due to autoinhibition of BicD, as is implied in the text, or simply because BICD2 CC1 is from another vertebrate species, whereas BicD is from insects? A more valid comparison would be between Drosophila full-length BicD and Drosophila BicD CC1.The revised manuscript includes a characterization of truncated *Drosophila* BicD^CC1^. See response to point (2) above.4) In Figure 1, the controls for unspecific binding of BICD2 CC1 and BicD to microtubules should be shown.

It is shown in Figure 1B of the revised manuscript.

5) In Figure 1, what do the error bars in Figure 1D represent?

The error bars in Figure 1C,D represent the standard error of run frequency from 5-15 microscopic fields. This is included in the figure legend.

7) In the context of the discussion of Figure 2, the authors mention pull downs were performed with CC1 vs CC2-CC3. These should be shown.

They are now included as Figure 2F.

8) The protein gel of the Egl preparation belongs to Figure 3.

Gels of Egl preparations are now Figure 3—figure supplement 1.

10) In Figure 4, there is a 2-fold drop in recruitment to microtubules between RNPs containing K10 mRNA and K10 mRNA without the transport/localization sequence (TLS). The authors speculate that regions in K10 mRNA other than the TLS could contribute to Egl binding. Another possibility is that any mRNA can un-specifically interact with Egl in this assay. Although the authors mention that tRNAs do not activate DDBE complexes, a better control would be to use another mRNA or a mix of mRNAs.

We have added two additional heterologous mRNAs (*S. cerevisiae ASH1* mRNA and β-actin mRNA) to Figure 4C of the revised manuscript. The frequency of these two mRNAs is very similar to *K10* mRNA lacking a zip code. This observation agrees with published work showing that in *Drosophila* embryos even non-localizing mRNAs associate with dynein and move bidirectionally (Amrute-Nayak and Bullock, 2012).

11) In Figure 4D: shifting the two channels relative to each other in the merged image is not appropriate; better to show the channels in separate.

Figure 4E now shows the separate and merged channels.

12) LC8 data:- why not present the results in graphs as done for Figures 5 – 7?- the authors state that they purified dynein without LC8. The protein gel of this preparation should be shown.

Figure 8 of the revised manuscript now shows speed and run length data for a head-to-head comparison of dynein expressed with or without LC8. The data presented were obtained with complexes reconstituted with 2 dyneins per dynactin. The gels of the dynein preparations are shown in Figure 8—figure supplement 1.

13) In Figure 5, as in Figure 1C,D, human BICD2 CC1 is compared with Drosophila full-length BicD, which is not ideal.

See responses to comments 1) and 2). We have now included new data obtained with truncated *Drosophila* BicD^CC1^ in Figure 1 and Figure 5.

20) What is the stoichiometry of the mRNA in the fast, highly processive transport complexes? This seems like an important aspect of the model that was not addressed experimentally. The authors could perform an experiment analogous to those done with dynein and Egl with mRNAs labeled with two different dyes.

New two color mRNA experiments are now included in Figure 9. The results show that 89% of moving complexes contain 1 mRNA. This result lead us to express an additional construct, a truncated Egl followed by a leucine zipper, which was able to activate dynein-dynactin, full-length BicD in the absence of mRNA (Figure 10). Results from these two sets of experiments allowed us to provide a mechanistic explanation for the role of mRNA in activation of dynein-dynactin-BicD motility.

21) The cartoons of the transport complexes are disappointing as they reflect so little of our structural understanding of these complexes. Conveying realistic spatial relationships is important for broader audiences who are less familiar with these complex machines. A few of the issues are: (1) Dynactin is shown as a rounded rectangle that represents neither its structure nor its size relative to dynein. This is particularly surprising when one of the authors of the paper is the person who discovered dynactin and was involved in two nice papers on the structure of DDB complexes. (2) Dynein is shown with 2-fold symmetry, which we now know is precisely what dynactin breaks to impose translational symmetry and activate the motor. (3) Coiled-coils are represented as globular domains, while they should look more like dynein's stalk (the only coiled-coil shown as such) or p150. (4) CC1 of BicD should be sandwiched between dynein and dynactin, reflecting its activating role; the figure shows it interacting only with dynactin.

Our cartoons have been improved to address the above comments.